# Follow-ups Also Matter: Improving Contextual Bandits via Post-serving Contexts

Chaoqi Wang [1]  Ziyu Ye [1]  Zhe Feng [2]  Ashwinkumar Badanidiyuru [2]  Haifeng Xu [1]

## Abstract

Standard contextual bandit problem assumes that all the relevant contexts are observed before the algorithm chooses an arm. This modeling paradigm, while useful, often falls short when dealing with problems in which valuable additional context can be observed after arm selection. For example, content recommendation platforms like Youtube, Instagram, Tiktok also observe valuable follow-up information pertinent to the user's reward after recommendation (e.g., how long the user stayed, what is the user's watch speed, etc.). To improve online learning efficiency in these applications, we study a novel contextual bandit problem with post-serving contexts and design a new algorithm, poLinUCB, that achieves tight regret under standard assumptions. Core to our technical proof is a robustified and generalized version of the well-known Elliptical Potential Lemma (EPL), which can accommodate noise in data. Such robustification is necessary for tackling our problem, and we believe it could also be of general interest. Extensive empirical tests on both synthetic and real-world datasets demonstrate the significant benefit of utilizing post-serving contexts as well as the superior performance of our algorithm over the state-of-the-art approaches.

## 1. Introduction

Contextual bandits represent a fundamental mathematical model that is employed across a variety of applications, such as personalized recommendations (Li et al., 2010; Wu et al., 2016) and online advertising (Schwartz et al., 2017; Nuara et al., 2018). In their conventional setup, at each round $t$, a learner observes the context $\boldsymbol{x}_t$, selects an arm $a_t \in \mathcal{A}$, and subsequently, observes its associated reward $r_{t,a_t}$. Despite being a basic and influential framework, it may not always

capture the complexity of real-world scenarios (Wang et al., 2016; Yang et al., 2020). Specifically, the learner often observes valuable follow-up information pertinent to the payoff post arm selection (henceforth, the *post-serving context*). Standard contextual bandits framework that neglects such post-serving contexts may result in significantly suboptimal performance due to model misspecification.

Consider an algorithm designed to recommend educational resources to a user by utilizing the user's partially completed coursework, interests, and proficiency as pre-serving context (exemplified in platforms such as Coursera). After completing the recommendation, the system can refine the user's profile by incorporating many post-serving context features such as course completion status, how much time spent on different educational resources, performances, etc. This transition naturally delineates a mapping from user attributes (i.e., the pre-serving context) to user's learning experiences and outcomes (i.e., the post-serving context). It is not difficult to see that similar scenarios happen in many other recommender system applications. For instance, in e-commerce platforms (e.g., Amazon, Etsy or any retailing website), the system will first recommend products based on the user's profile information, purchasing pattern and browsing history, etc.; post recommendations, the system can then update these information by integrating post-serving contexts such as this recent purchase behaviour and product reviews. Similarly, media content recommendation platforms like Youtube, Instagram and Tiktok, also observe many post-serving features (e.g., how long the user stayed) that can refine the system's estimation about users' interaction behavior as well as the rewards.

A common salient point in all the aforementioned scenarios is that the post-serving context are prevalent in many recommender systems; moreover, despite being unseen during the recommendation/serving phase, they can be estimated from the pre-serving context given enough past data. More formally, we assume that there exists a learnable mapping $\phi^\star(\cdot) : \mathbb{R}^{d_x} \to \mathbb{R}^{d_z}$ that maps pre-serving feature $\boldsymbol{x} \in \mathbb{R}^{d_x}$ to the expectation of the post-serving feature $\boldsymbol{z} \in \mathbb{R}^{d_z}$, i.e., $\mathbb{E}[\boldsymbol{z}|\boldsymbol{x}] = \phi^\star(\boldsymbol{x})$.

Unsurprisingly, and as we will also show, integrating the estimation of such post-serving features can significantly help

---

[1]University of Chicago [2]Google Research. Correspondence to: Chaoqi Wang <chaoqi@uchicago.edu>.

Interactive Learning with Implicit Human Feedback Workshop at ICML 2023, Honolulu, Hawaii, USA.

to enhance the performance of contextual bandits. However, most of the existing contextual bandit algorithms, e.g., (Auer, 2002; Li et al., 2010; Chu et al., 2011; Agarwal et al., 2014; Tewari and Murphy, 2017), are not designed to accommodate the situations with post-serving contexts. We observe that directly applying these algorithms by ignoring post-serving contexts may lead to linear regret, whereas simple modification of these algorithms will also be sub-optimal. To address these shortcomings, this work introduces a novel algorithm, poLinUCB. Our algorithm leverages historical data to simultaneously estimate reward parameters and the functional mapping from the pre- to post-serving contexts so to optimize arm selection and achieves sublinear regret. En route to analyzing our algorithm, we also developed new technique tools that may be of independent interest.

**Main Contributions.**

- First, we introduce a new family of contextual linear bandit problems. In this framework, the decision-making process can effectively integrate post-serving contexts, premised on the assumption that the expectation of post-serving context as a function of the pre-serving context can be gradually learned from historical data. This new model allows us to develop more effective learning algorithms in many natural applications with post-serving contexts.

- Second, to study this new model, we developed a robustified and generalized version of the well-regarded elliptical potential lemma (EPL) in order to accommodate random noise in the post-serving contexts. While this generalized EPL is an instrumental tool in our algorithmic study, we believe it is also of independent interest due to the broad applicability of EPL in online learning.

- Third, building upon the generalized EPL, we design a new algorithm poLinUCB and prove that it enjoys a regret bound $\widetilde{\mathcal{O}}(T^{1-\alpha}d_u^\alpha + d_u\sqrt{TK})$, where $T$ denotes the time horizon and $\alpha \in [0, 1/2]$ is the learning speed of the pre- to post-context mapping function $\phi^\star(\cdot)$, whereas $d_u = d_x + d_z$ and $K$ denote the parameter dimension and number of arms. When $\phi^\star(\cdot)$ is easy to learn, e.g., $\alpha = 1/2$, the regret bound becomes $\widetilde{\mathcal{O}}(\sqrt{Td_u} + d_u\sqrt{TK})$ and is tight. For general functions $\phi^\star(\cdot)$ that satisfy $\alpha \leq 1/2$, this regret bound degrades gracefully as the function becomes more difficult to learn, i.e., as $\alpha$ decreases.

- Lastly, we empirically validate our proposed algorithm through thorough numerical experiments on both simulated benchmarks and real-world datasets. The results demonstrate that our algorithm surpasses existing state-of-the-art solutions. Furthermore, they highlight the tangible benefits of incorporating the functional relationship between pre- and post-serving contexts into the model, thereby affirming the effectiveness of our modeling.

**Related Works.** Our work lies in the extensive linear contextual bandits literature, however, most of existing studies assume the full contexts are observable before playing actions (Abe et al., 2003; Auer, 2002; Dani et al., 2008; Rusmevichientong and Tsitsiklis, 2010; Lu et al., 2010; Filippi et al., 2010; Li et al., 2010; Chu et al., 2011; Abbasi-Yadkori et al., 2011; Li et al., 2017; Jun et al., 2017) and they demonstrate the use of upper confidence bounds for balancing exploration and exploitation, proving minimax optimal regret bounds using the confidence ellipsoids and the elliptical potential lemma (EPL). In sharp contrast, we assume partially observable context in this paper, where only partial contexts are observed before making decisions. Partial information contextual bandits, though limited, have been studied (Wang et al., 2016; Qi et al., 2018; Yang et al., 2020; Park and Faradonbeh, 2021; Yang and Ren, 2021; Zhu and Kveton, 2022), focusing on predicting context information through context history analysis or selective expert requests. Differently, our work introduces a novel problem setting, separating contexts into pre-serving and post-serving categories. To achieve near-optimal regret bound in this setting, we propose a generalized elliptical potential lemma to handle the additional noise introduced by the post-serving features. While the EPL (Lai and Wei, 1982) and its generalizations have been widely used in stochastic linear bandit problems (Auer, 2002; Dani et al., 2008; Chu et al., 2011; Abbasi-Yadkori et al., 2011; Li et al., 2019; Zhou et al., 2020; Wang et al., 2022; Carpentier et al., 2020; Hamidi and Bayati, 2022), they fall short for the analysis in our context. See Appendix A.1 for further related works.

## 2. Linear Bandits with Post-serving Contexts

**Basic setup.** We hereby delineate a basic setup of linear contextual bandits within the scope of the partial information setting, whereas multiple generalizations of our framework can be found in Section 5. This setting involves a finite and discrete action space, represented as $\mathcal{A} = [K]$. Departing from the classic contextual bandit setup, the context in our model is bifurcated into two distinct components: the *pre-serving* context, denoted as $\boldsymbol{x} \in \mathbb{R}^{d_x}$, and the *post-serving* context, signified as $\boldsymbol{z} \in \mathbb{R}^{d_z}$. When it is clear from context, we sometimes refer to pre-serving context simply as *context* as in classic setup, but always retain the post-serving context notion to emphasize its difference. We will denote $\boldsymbol{X}_t = \sum_{s=1}^t \boldsymbol{x}_s \boldsymbol{x}_s^\top + \lambda \boldsymbol{I}$ and $\boldsymbol{Z}_t = \sum_{s=1}^t \boldsymbol{z}_s \boldsymbol{z}_s^\top + \lambda \boldsymbol{I}$. For the sake of brevity, we employ $\boldsymbol{u} = (\boldsymbol{x}, \boldsymbol{z})$ to symbolize the stacked vector of $\boldsymbol{x}$ and $\boldsymbol{z}$, with $d_u = d_x + d_z$ and $\|\boldsymbol{u}\|_2 \leq L_u$. The pre-serving context is available during arm selection, while the post-serving context is disclosed *post* the arm selection. For each arm $a \in \mathcal{A}$, the payoff, $r_a(\boldsymbol{x}, \boldsymbol{z})$, is delineated as follows:

$$r_a(\boldsymbol{x}, \boldsymbol{z}) = \boldsymbol{x}^\top \boldsymbol{\theta}_a^\star + \boldsymbol{z}^\top \boldsymbol{\beta}_a^\star + \eta,$$

where $\boldsymbol{\theta}_a^\star$ and $\boldsymbol{\beta}_a^\star$ represent the parameters associated with the arm, unknown to the learner, whereas $\eta$ is a random noise sampled from an $R_\eta$-sub-Gaussian distribution. We use $\|\boldsymbol{x}\|_p$ to denote the $p$-norm of a vector $\boldsymbol{x}$, and $\|\boldsymbol{x}\|_{\boldsymbol{A}} := \sqrt{\boldsymbol{x}^\top \boldsymbol{A} \boldsymbol{x}}$ is the matrix norm. For convenience, we assume $\|\boldsymbol{\theta}_a^\star\|_2 \le 1$ and $\|\boldsymbol{\beta}_a^\star\|_2 \le 1$ for all $a \in \mathcal{A}$. Additionally, we posit that the norm of the pre-serving and post-serving contexts satisfies $\|\boldsymbol{x}\|_2 \le L_x$ and $\|\boldsymbol{z}\|_2 \le L_z$, respectively, and $\max_{t \in [T]} \sup_{a,b \in \mathcal{A}} \langle \boldsymbol{\theta}_a^\star - \boldsymbol{\theta}_b^\star, \boldsymbol{x}_t \rangle \le 1$ and $\max_{t \in [T]} \sup_{a,b \in \mathcal{A}} \langle \boldsymbol{\beta}_a^\star - \boldsymbol{\beta}_b^\star, \boldsymbol{z}_t \rangle \le 1$, same as in (Lattimore and Szepesvári, 2020).

### 2.1. Problem Settings and Assumptions.

The learning process proceeds as follows at each time step $t = 1, 2, \cdots, T$:

1. The learner observes the context $\boldsymbol{x}_t$.
2. An arm $a_t \in [K]$ is selected by the learner.
3. The learner observes the realized reward $r_{t,a_t}$ and the post-serving context, $\boldsymbol{z}_t$.

Without incorporating the post-serving context, one may incur linear regret as a result of model misspecification, as illustrated in the following observation. To see this, consider a setup with two arms, $a_1$ and $a_2$, and a context $x \in \mathbb{R}$ drawn uniformly from the set $\{-3, -1, 1\}$ with $\phi^\star(x) = x^2$. The reward functions for the arms are noiseless and determined as $r_{a_1}(x) = x + x^2/2$ and $r_{a_2}(x) = -x - x^2/2$. It can be observed that $r_{a_1}(x) > r_{a_2}(x)$ when $x \in \{-3, 1\}$ and $r_{a_1}(x) < r_{a_2}(x)$ when $x = -1$. Any linear bandit algorithm that solely dependent on the context $x$ (ignoring $\phi^\star(x)$) will inevitably suffer from linear regret, since it is impossible to have a linear function (i.e., $r(x) = \theta x$) that satisfies the above two inequalities simultaneously.

**Observation 1.** *There exists linear bandit environments in which any online algorithm without using post-serving context information will have $\Omega(T)$ regret.*

Therefore, it is imperative that an effective learning algorithm must leverage the post-serving context, denoted as $\boldsymbol{z}_t$. As one might anticipate, in the absence of any relationship between $\boldsymbol{z}_t$ and $\boldsymbol{x}_t$, it would be unfeasible to extrapolate any information regarding $\boldsymbol{z}_t$ while deciding which arm to pull, a point at which only $\boldsymbol{x}_t$ is known. Consequently, it is reasonable to hypothesize a correlation between $\boldsymbol{z}_t$ and $\boldsymbol{x}_t$. This relationship is codified in the subsequent learnability assumption.

Specifically, we make the following natural assumption — there exists an algorithm that can learn the mean of the post-serving context $\boldsymbol{z}_t$, conditioned on $\boldsymbol{x}_t$. Our analysis will be general enough to accommodate different convergence rates of the learning algorithm, as one would naturally expect,

the corresponding regret will degrade as this learning algorithm's convergence rate becomes worse. More specifically, we posit that, given the context $\boldsymbol{x}_t$, the post-serving context $\boldsymbol{z}_t$ is generated as[1]

$$\boldsymbol{z}_t = \phi^\star(\boldsymbol{x}_t) + \boldsymbol{\epsilon}_t, \ i.e., \ \phi^\star(\boldsymbol{x}) = \mathbb{E}[\boldsymbol{z}|\boldsymbol{x}].$$

Here, $\boldsymbol{\epsilon}_t$ is a zero-mean noise vector in $\mathbb{R}^{d_z}$, and $\phi^\star : \mathbb{R}^{d_x} \to \mathbb{R}^{d_z}$ can be viewed as the post-serving context generating function, which is unknown to the learner. However, we assume $\phi^\star$ is learnable in the following sense.

**Assumption 1** (Generalized learnability of $\phi^*$)**.** *There exists an algorithm that, given $t$ pairs of examples $\{(\boldsymbol{x}_s, \boldsymbol{z}_s)\}_{s=1}^t$ with arbitrarily chosen $\boldsymbol{x}_s$'s, outputs an estimated function of $\phi^\star : \mathbb{R}^{d_x} \to \mathbb{R}^{d_z}$ such that for any $\boldsymbol{x} \in \mathbb{R}^{d_x}$, the following holds with probability at least $1 - \delta$,*

$$e_t^\delta := \left\| \widehat{\phi}_t(\boldsymbol{x}) - \phi^\star(\boldsymbol{x}) \right\|_2 \le C_0 \cdot \left( \|\boldsymbol{x}\|_{\boldsymbol{X}_t^{-1}}^2 \right)^\alpha \cdot \log(t/\delta),$$

*where $\alpha \in (0, 1/2]$ and $C_0$ is some universal constant.*

The aforementioned assumption encompasses a wide range of learning scenarios, each with different rates of convergence. Generally, the value of $\alpha$ is directly proportional to the speed of learning; the larger the value of $\alpha$, the quicker the learning rate. Later, we will demonstrate that the regret of our algorithm is proportional to $O(T^{1-\alpha})$, exhibiting a graceful degradation as $\alpha$ decreases. The ensuing proposition demonstrates that for linear functions, $\alpha = 1/2$. This represents the best learning rate that can be accommodated[2]. In this scenario, the regret of our algorithm is $O(\sqrt{T})$, aligning with the situation devoid of post-serving contexts (Li et al., 2010; Abbasi-Yadkori et al., 2011).

**Observation 2.** *Suppose $\phi(\cdot)$ is a linear function, i.e., $\phi(\boldsymbol{x}) = \boldsymbol{\Phi}^\top \boldsymbol{x}$ for some $\boldsymbol{\Phi} \in \mathbb{R}^{d_x \times d_z}$, then $e_t^\delta = \mathcal{O}\left(\|\boldsymbol{x}\|_{\boldsymbol{X}_t^{-1}} \cdot \log(t/\delta)\right)$.*

This observation follows from the following inequalities $\|\phi_t(\boldsymbol{x}) - \phi^\star(\boldsymbol{x})\| = \|\widehat{\boldsymbol{\Phi}}_t^\top \boldsymbol{x} - \boldsymbol{\Phi}^{\star\top}\boldsymbol{x}\| \le \|\widehat{\boldsymbol{\Phi}}_t - \boldsymbol{\Phi}^\star\|_{\boldsymbol{X}_t} \cdot \|\boldsymbol{x}\|_{\boldsymbol{X}_t^{-1}} = \mathcal{O}(\|\boldsymbol{x}\|_{\boldsymbol{X}_t^{-1}} \cdot \log(\frac{t}{\delta}))$, where the last equation is due to the confidence ellipsoid bound (Abbasi-Yadkori et al., 2011).

---

[1]Given this, an alternative view of $r_a$ is to treat it as a function of $\boldsymbol{x}$ as follows: $r_a(\boldsymbol{x}) = \boldsymbol{x}^\top \boldsymbol{\theta}_a^\star + \phi^\star(\boldsymbol{x})^\top \boldsymbol{\beta}_a^\star + \text{noise}$. Our algorithm can be viewed as a two-phase learning of this structured function, i.e., using $(\boldsymbol{x}_t, \boldsymbol{z}_t)$ to learn $\phi^\star$ that is shared among all arms and then using learned $\hat{\phi}$ to estimate each arm $a$'s reward parameters.

[2]To be precise, our analysis can extend to instances where $\alpha > 1/2$. However, this would not enhance the regret bound since the regret is already $\Omega(\sqrt{T})$, even with knowledge of $\boldsymbol{z}$. This would only further complicate our notation, and therefore, such situations are not explicitly examined in this paper.

## 2.2. Warm-up: Why Natural Attempts May Be Inadequate?

Given the learnability assumption of $\phi^\star$, one natural idea for solving the above problem is to estimate $\phi^\star$, and then run the standard LinUCB algorithm to estimate $(\boldsymbol{\theta}_a, \boldsymbol{\beta}_a)$ together by treating $(\boldsymbol{x}_t, \widehat{\phi}_t(\boldsymbol{x}_t))$ as the true contexts. Indeed, this is the approach adopted by Wang et al. (2016) for addressing a similar problem of missing contexts $\boldsymbol{z}_t$, except that they used a different unsurprised-learning-based approach to estimate the context $\boldsymbol{z}_t$ due to not being able to observing any data about $\boldsymbol{z}_t$. Given the estimation of $\widehat{\phi}$, their algorithm — which we term it as *LinUCB* $(\widehat{\phi})$ — iteratively carries out the steps below at each iteration $t$ (see Appendix 2 for additional details): 1) Estimation of the context-generating function $\widehat{\phi}_t(\cdot)$ from historical data; 2) Solve of the following regularized least square problem for each arm $a \in \mathcal{A}$, with regularization coefficient $\lambda \geq 0$:

$$\ell_t(\boldsymbol{\theta}_a, \boldsymbol{\beta}_a) = \sum_{s \in [t]: a_s = a} \left( r_{s,a} - \boldsymbol{x}_t^\top \boldsymbol{\theta}_a - \widehat{\phi}_s(\boldsymbol{x}_s)^\top \boldsymbol{\beta}_a \right)^2$$
$$+ \lambda \left( \|\boldsymbol{\theta}_a\|_2^2 + \|\boldsymbol{\beta}_a\|_2^2 \right), \tag{1}$$

Under the assumption that the initialized parameters in their estimations are very close to the global optimum, Wang et al. (2016) were able to show the $O(\sqrt{T})$ regret of this algorithm. However, it turns out that this algorithm will fail to yield an satisfying regret bound without their strong assumption on very close parameter initialization, because the errors arising from $\widehat{\phi}(\cdot)$ will significantly enlarge the confidence set of $\widehat{\boldsymbol{\theta}}_a$ and $\widehat{\boldsymbol{\beta}}_a$.[3] Thus after removing their initialization assumption, the best possible regret bound we can possibly achieve is of order $\widetilde{\mathcal{O}}(T^{3/4})$, as illustrated in the subsequent proposition.

**Proposition 1** (Regret of LinUCB-$(\widehat{\phi})$)**.** *The regret of LinUCB-$(\widehat{\phi})$ in Algorithm 2 is upper bounded by $\widetilde{\mathcal{O}}\left( T^{1-\alpha} d_u^\alpha + T^{1-\alpha/2} \sqrt{K d_u^{1+\alpha}} \right)$ with probability at least $1 - \delta$, by carefully setting the regularization coefficient $\lambda = \Theta(L_u d_u^\alpha T^{1-\alpha} \log (T/\delta))$ in Equation 1.*

Since $\alpha \in [0, 1/2]$, the best possible regret upper bound above is $\widetilde{\mathcal{O}}(T^{3/4})$, which is considerably inferior to the sought-after regret bound of $\widetilde{\mathcal{O}}(\sqrt{T})$. Such deficiency of LinUCB-$(\widehat{\phi})$ is further observed in all our experiments in Section 6 as well. These motivate our following design of a new online learning algorithm to address the challenge of

---

[3]Specifically, Wang et al. (2016) assume that the initial estimation of $\widehat{\phi}(\cdot)$ is already very close to $\widehat{\phi}^*(\cdot)$ such that the error arising from $\widehat{\phi}(\cdot)$ diminishes exponentially fast due to the local convergence property of alternating least squares algorithm (Uschmajew, 2012). This strong assumption avoids significant expansion of the confidence sets, but is less realistic in applications so we do not impose such assumption.

post-serving context, during which we also developed a new technical tool which may be of independent interest to the research community.

# 3. A Robustified and Generalized Elliptical Potential Lemma

It turns out that solving the learning problem above requires some novel designs; core to these novelties is a robustified and generalized version of the well-known elliptical potential lemma (EPL), which may be of independent interest. This widely used lemma states a fact about a sequence of vectors $\boldsymbol{x}_1, \cdots, \boldsymbol{x}_T \in \mathbb{R}^d$. Intuitively, it captures the rate of the sum of additional information contained in each $\boldsymbol{x}_t$, relative to its predecessors $\boldsymbol{x}_1, \cdots, \boldsymbol{x}_{t-1}$. Formally,

**Lemma** (Original Elliptical Potential Lemma)**.** *Suppose (1) $\boldsymbol{X}_0 \in \mathbb{R}^{d \times d}$ is any positive definite matrix; (2) $\boldsymbol{x}_1, \ldots, \boldsymbol{x}_T \in \mathbb{R}^d$ is any sequence of vectors; and (3) $\boldsymbol{X}_t = \boldsymbol{X}_0 + \sum_{s=1}^t \boldsymbol{x}_s \boldsymbol{x}_s^\top$. Then the following inequality holds*

$$\sum_{t=1}^T 1 \wedge \|\boldsymbol{x}_t\|_{\boldsymbol{X}_{t-1}^{-1}}^2 \leq 2 \log \left( \frac{\det \boldsymbol{X}_T}{\det \boldsymbol{X}_0} \right),$$

*where $a \wedge b = \min\{a, b\}$ is the min among $a, b \in \mathbb{R}$.*

To address our new contextual bandit setup with post-serving contexts, it turns out that we will need to robustify and generlaize the above lemma to accommodate noises in $\boldsymbol{x}_t$ vectors and slower learning rates. Specifically, we present the following variant of the EPL lemma.

**Lemma 1** (Generalized Elliptical Potential Lemma)**.** *Suppose (1) $\boldsymbol{X}_0 \in \mathbb{R}^{d \times d}$ is any positive definite matrix; (2) $\boldsymbol{x}_1, \ldots, \boldsymbol{x}_T \in \mathbb{R}^d$ is a sequence of vectors with bounded $l_2$ norm $\max_t \|\boldsymbol{x}_t\| \leq L_x$; (3) $\boldsymbol{\epsilon}_1, \ldots, \boldsymbol{\epsilon}_T \in \mathbb{R}^d$ is a sequence of independent (not necessarily identical) bounded zero-mean noises satisfying $\max_t \|\boldsymbol{\epsilon}_t\| \leq L_\epsilon$ and $\mathbb{E}[\boldsymbol{\epsilon}_t \boldsymbol{\epsilon}_t^\top] \succcurlyeq \sigma_\epsilon^2 \boldsymbol{I}$ for any $t$; and (4) $\widetilde{\boldsymbol{X}}_t$ is defined as follows:*

$$\widetilde{\boldsymbol{X}}_t = \boldsymbol{X}_0 + \sum_{s=1}^t (\boldsymbol{x}_s + \boldsymbol{\epsilon}_s)(\boldsymbol{x}_s + \boldsymbol{\epsilon}_s)^\top \in \mathbb{R}^{d \times d}.$$

*Then, for any $p \in [0, 1]$, the following inequality holds with probability at least $1 - \delta$,*

$$\sum_{t=1}^T \left( 1 \wedge \|\boldsymbol{x}_t\|_{\widetilde{\boldsymbol{X}}_{t-1}^{-1}}^2 \right)^p \leq 2^p T^{1-p} \log^p \left( \frac{\det \boldsymbol{X}_T}{\det \boldsymbol{X}_0} \right)$$
$$+ \frac{8 L_\epsilon^2 (L_\epsilon + L_x)^2}{\sigma_\epsilon^4} \log \left( \frac{32 d L_\epsilon^2 (L_\epsilon + L_x)^2}{\delta \sigma_\epsilon^4} \right) \tag{2}$$

Note that the second term is independent of time horizon $T$ and only depends on the setup parameters. Generally, this can be treated as a constant. Before describing main proof idea of the lemma, we make a few remarks regarding

Lemma 1 to highlight the significance of these generalizations.

1. The original Ellipsoid Prediction Lemma (EPL) corresponds to the specific case of $p = 1$, while Lemma 1 is applicable for any $p \in [0, 1]$. Notably, the $(1 - p)$ rate in the $T^{1-p}$ term of Inequality 2 is tight for *every* $p$. In fact, this rate is tight even for $\boldsymbol{x}_t = 1 \in \mathbb{R}, \forall t$ and $\boldsymbol{X}_0 = 1 \in \mathbb{R}$ since, under these conditions, $\|\boldsymbol{x}_t\|^2_{\boldsymbol{X}_{t-1}^{-1}} = 1/t$ and, consequently, $\sum_{t=1}^T \left( 1 \wedge \|\boldsymbol{x}_t\|^2_{\boldsymbol{X}_{t-1}^{-1}} \right)^p = \sum_{t=1}^T t^{-p}$, yielding a rate of $T^{1-p}$. This additional flexibility gained by allowing a general $p \in [0, 1]$ (with the original EPL corresponding to $p = 1$) helps us to accommodate slower convergence rates when learning the mean context from observed noisy contexts, as formalized in Assumption 1.

2. A crucial distinction between Lemma 1 and the original EPL lies in the definition of the noisy data matrix $\widetilde{\boldsymbol{X}}_t$ in Equation 1, which permits noise. However, the measured context vector $\boldsymbol{x}_t$ does *not* have noise. This is beneficial in scenarios where a learner observes noisy contexts but seeks to establish an upper bound on the prediction error based on the underlying noise-free context or the mean context. Such situations are not rare in real applications; our problem of contextual bandits with post-serving contexts is precisely one of such case — while choosing an arm, we can estimate the mean post-serving context conditioned on the observable pre-serving context but are only able to observe the noisy realization of post-serving contexts after acting.

3. Other generalized variants of the EPL have been recently proposed and found to be useful in different contexts. For instance, (Carpentier et al., 2020) extends the EPL to allow for the $\boldsymbol{X}_t^{-p}$-norm, as opposed to the $\boldsymbol{X}_t^{-1}$-norm, while (Hamidi and Bayati, 2022) explores a generalized form of the $1 \wedge \|\varphi(\boldsymbol{x}_t)\|^2_{\boldsymbol{X}_{t-1}^{-1}}$ term, which is motivated by variance reduction in non-Gaussian linear regression models. Nevertheless, to the best of our knowledge, our generalized version is novel and has not been identified in prior works.

*Proof Sketche of Lemma 1.* The formal proof of this lemma is involved and deferred to Appendix C.1. At a high level, our proof follows procedure for proving the original EPL. However, to accommodate the noises in the data matrix, we have to introduce new matrix concentration tools to the original (primarily algebraic) proof, and also identify the right conditions for the argument to go through. A key lemma to our proof is a high probability bound regarding the constructed noisy data matrix $\widetilde{\boldsymbol{X}}_t$ (Lemma 2 in Appendix C.1) that we derive based on Bernstein's Inequality for matrices under spectral norm (Tropp et al., 2015). We prove that, under mild assumptions on the noise, $\|\boldsymbol{x}_t\|^2_{\widetilde{\boldsymbol{X}}_{t-1}^{-1}} \leq \|\boldsymbol{x}_t\|^2_{\boldsymbol{X}_{t-1}^{-1}}$ with high probability for any $t$. Next, we have to apply the

union bound and this lemma to show that the above matrix inequality holds for *every* $t \geq 1$ with high probability. Unfortunately, this turns out to not be true because when $t$ is very small (e.g., $t = 1$), the above inequality cannot hold with high probability. Therefore, we have to use the union bound in a carefully tailored way by excluding all $t$'s that are smaller than a certain threshold (chosen optimally by solving certain inequalities) and handling these terms with small $t$ separately (which is the reason of the second $\mathcal{O}(\log(1/\delta))$ term in Inequality 2). Finally, we refine the analysis of sthe standard EPL by allowing the exponent $p$ in $(1 \wedge \|\boldsymbol{x}_t\|^2_{\widetilde{\boldsymbol{X}}_{t-1}^{-1}})^p$ and derive an upper bound on the sum $\sum_{t=1}^T (1 \wedge \|\boldsymbol{x}_t\|^2_{\widetilde{\boldsymbol{X}}_{t-1}^{-1}})^p$ with high probability. These together yeilds a robustified and generalized version of EPL as in Lemma 1. □

## 4. No Regret Learning in Linear Bandits with Post-Serving Contexts

### 4.1. The Main Algorithm

In the ensuing section, we introduce our algorithm, poLin-UCB, designed to enhance linear contextual bandit learning through the incorporation of post-serving contexts and address the issue arose from the algorithm introduced in Section 2.2. The corresponding pseudo-code is delineated in Algorithm 1. Unlike the traditional LinUCB algorithm, which solely learns and sustains confidence sets for parameters (i.e., $\widehat{\boldsymbol{\beta}}_a$ and $\widehat{\boldsymbol{\theta}}_a$ for each $a$), our algorithm also *simultaneously* manages the same for the post-serving context generating function, $\widehat{\phi}(\cdot)$. Below, we expound on our methodology for parameter learning and confidence set construction.

**Parameter learning**. During each iteration $t$, we fit the function $\widehat{\phi}_t(\cdot)$ and the parameters $\{\widehat{\boldsymbol{\theta}}_{t,a}\}_{a \in \mathcal{A}}$ and $\{\widehat{\boldsymbol{\beta}}_{t,a}\}_{a \in \mathcal{A}}$. To fit $\widehat{\phi}_t(\cdot)$, resort to the conventional empirical risk minimization (ERM) framework. As for $\{\widehat{\boldsymbol{\theta}}_{t,a}\}_{a \in \mathcal{A}}$ and $\{\widehat{\boldsymbol{\beta}}_{t,a}\}_{a \in \mathcal{A}}$, we solve the following least squared problem for each arm $a$,

$$\ell_t(\boldsymbol{\theta}_a, \boldsymbol{\beta}_a) = \sum_{s \in [t]: a_s = a} \left( r_{s,a} - \boldsymbol{x}_s^\top \boldsymbol{\theta}_a - \boldsymbol{z}_s^\top \boldsymbol{\beta}_a \right)^2$$
$$+ \lambda \left( \|\boldsymbol{\theta}_a\|_2^2 + \|\boldsymbol{\beta}_a\|_2^2 \right). \quad (3)$$

For convenience, we use $\boldsymbol{w}$ and $\boldsymbol{u}$ to denote $(\boldsymbol{\theta}, \boldsymbol{\beta})$ and $(\boldsymbol{x}, \boldsymbol{z})$ respectively. The closed-form solutions to $\widehat{\boldsymbol{\theta}}_{t,a}$ and $\widehat{\boldsymbol{\beta}}_{t,a}$ for each arm $a \in \mathcal{A}$ are

$$\widehat{\boldsymbol{w}}_{t,a} := \begin{bmatrix} \widehat{\boldsymbol{\theta}}_{t,a} \\ \widehat{\boldsymbol{\beta}}_{t,a} \end{bmatrix} = \boldsymbol{A}_{t,a}^{-1} \boldsymbol{b}_{t,a} \quad (4)$$

$\boldsymbol{A}_{t,a} = \lambda \boldsymbol{I} + \sum_{s:a_s=a}^t \boldsymbol{u}_s \boldsymbol{u}_s^\top$ and $\boldsymbol{b}_{t,a} = \sum_{s:a_s=a}^t r_{s,a} \boldsymbol{u}_s$.

**Algorithm 1** poLinUCB (*Linear UCB with post-serving contexts*)

1: **for** $t = 0, 1, \ldots, T$ **do**
2:     Receive the *pre-serving context* $\boldsymbol{x}_t$
3:     Compute the optimistic parameters by maximizing the UCB objective

$$\left(a_t, \widetilde{\phi}_t(\boldsymbol{x}_t), \widetilde{\boldsymbol{w}}_t\right) = \underset{(a, \phi, \boldsymbol{w}_a) \in [K] \times \mathcal{C}_{t-1}\left(\widehat{\phi}_{t-1}, \boldsymbol{x}_t\right) \times \mathcal{C}_{t-1}(\widehat{\boldsymbol{w}}_{t-1,a})}{\arg\max} \begin{bmatrix} \boldsymbol{x}_t \\ \phi(\boldsymbol{x}_t) \end{bmatrix}^\top \boldsymbol{w}_a.$$

4:     Play the arm $a_t$ and receive the realized *post-serving context* as $\boldsymbol{z}_t$ and the real-valued reward

$$r_{t,a_t} = \begin{bmatrix} \boldsymbol{x}_t \\ \boldsymbol{z}_t \end{bmatrix}^\top \boldsymbol{w}_{a_t}^\star + \eta_t.$$

5:     Compute $\widehat{\boldsymbol{w}}_{t,a}$ using Equation 3 for each $a \in \mathcal{A}$.
6:     Compute the estimated post-serving context generating function $\widehat{\phi}_t(\cdot)$ using ERM.
7:     Update confidence sets $\mathcal{C}_t(\widehat{\boldsymbol{w}}_{t,a})$ and $\mathcal{C}_t(\widehat{\phi}_t, \boldsymbol{x}_t)$ for each $a$ based on Equations 6 and 5.
8: **end for**

**Confidence set construction.** At iteration $t$, we construct the confidence set for $\widehat{\phi}_t(\boldsymbol{x}_t)$ by

$$\mathcal{C}_t\left(\widehat{\phi}_t, \boldsymbol{x}_t\right) := \left\{ \boldsymbol{z} \in \mathbb{R}^d : \left\|\widehat{\phi}_t(\boldsymbol{x}_t) - \boldsymbol{z}\right\|_2 \le e_t^\delta \right\}. \quad (5)$$

Similarly, we can construct the confidence set for the parameters $\widehat{\boldsymbol{w}}_{t,a}$ for each arm $a \in \mathcal{A}$ by

$$\mathcal{C}_t(\widehat{\boldsymbol{w}}_{t,a}) := \left\{ \boldsymbol{w} \in \mathbb{R}^{d_x + d_z} : \|\boldsymbol{w} - \widehat{\boldsymbol{w}}_{t,a}\|_{\boldsymbol{A}_{t,a}} \le \zeta_{t,a} \right\}, \quad (6)$$

where $\zeta_{t,a} = 2\sqrt{\lambda} + R_\eta \sqrt{d_u \log\left((1 + n_t(a)L_u^2/\lambda)/\delta\right)}$ and $n_t(a) = \sum_{s=1}^t \mathbb{1}[a_s = a]$. Additionally, we further define $\zeta_t := \max_{a \in \mathcal{A}} \zeta_{t,a}$. By the assumption 1 and Lemma 3, we have the followings hold with probability at least $1 - \delta$ for each of the following events,

$$\phi^\star(\boldsymbol{x}_t) \in \mathcal{C}_t\left(\widehat{\phi}_t, \boldsymbol{x}_t\right) \quad \text{and} \quad \boldsymbol{w}^\star \in \mathcal{C}_t(\widehat{\boldsymbol{w}}_{t,a}). \quad (7)$$

### 4.2. Regret Analysis

In the forthcoming section, we establish the regret bound. Our proof is predicated upon the conventional proof of Lin-UCB (Li et al., 2010) in conjunction with our robust elliptical potential lemma. The pseudo-regret (Audibert et al., 2009) within this partial contextual bandit problem is defined as,

$$R_T = \text{Regret}(T) = \sum_{t=1}^T \left(r_{t,a_t^\star} - r_{t,a_t}\right), \quad (8)$$

in which we reload the reward by ignoring the noise,

$$r_{t,a} = \langle \boldsymbol{\theta}_a^\star, \boldsymbol{x}_t \rangle + \langle \boldsymbol{\beta}_a^\star, \phi^\star(\boldsymbol{x}_t) \rangle, \quad (9)$$

where $a_t^\star = \arg\max_{a \in \mathcal{A}} \langle \boldsymbol{\theta}_a^\star, \boldsymbol{x}_t \rangle + \langle \boldsymbol{\beta}_a^\star, \phi^\star(\boldsymbol{x}_t) \rangle$. It is crucial to note that our definition of the optimal action, $a_t^\star$,

in Eq. 9 depends on $\phi^\star(\boldsymbol{x}_t)$ as opposed to $\boldsymbol{z}_t$. This dependency ensures a more pragmatic benchmark, as otherwise, the noise present in $\boldsymbol{z}$ would invariably lead to a linear regret, regardless of the algorithm implemented. In the ensuing section, we present our principal theoretical outcomes, which provide an upper bound on the regret of our poLinUCB algorithm.

**Theorem 1** (Regret of poLinUCB). *The regret of poLinUCB in Algorithm 1 is upper bounded by $\widetilde{\mathcal{O}}\left(T^{1-\alpha}d_u^\alpha + d_u\sqrt{TK}\right)$ with probability at least $1 - \delta$, if $T = \Omega(\log(1/\delta))$.*

The first term in the bound is implicated by learning the function $\phi^\star(\cdot)$. Conversely, the second term resembles the one derived in conventional contextual linear bandits, with the exception that our dependency on $d_u$ is linear. This linear dependency is a direct consequence of our generalized robust elliptical potential lemma. The proof is deferred in Appendix C.2.

## 5. Generalizations

So far we have focused on a basic linear bandit setup with post-serving features. Our results and analysis can be easily generalized to other variants of linear bandits, including those with feature mappings, and below we highlight some of these generalizations. They use similar proof ideas, up to some technical modifications; we thus defer all their formal proofs to Appendix C.3.

### 5.1. Generalization to Action-Dependent Contexts

Our basic setup in Section 2 has a single context $\boldsymbol{x}_t$ at any time step $t$. This can be generalized to action-dependent

contexts settings as studied in previous works (e.g., Li et al. (2010)). That is, during each iteration indexed by $t$, the learning algorithm observes a context $\boldsymbol{x}_{t,a}$ for each individual arm $a \in \mathcal{A}$. Upon executing the action of pulling arm $a_t$, the corresponding post-serving context $\boldsymbol{z}_{t,a_t}$ is subsequently revealed. Notwithstanding, the post-serving context for all alternative arms remains unobserved. The entire procedure is the same as that of Section 2.

In extending this framework, we persist in our assumption that for each arm $a \in \mathcal{A}$, there exists a specific function $\phi_a^\star(\cdot) : \mathbb{R}^{d_x} \to \mathbb{R}^{d_z}$ that generates the post-serving context $\boldsymbol{z}$ upon receiving $\boldsymbol{x}$ associated with arm $a \in \mathcal{A}$. The primary deviation from our preliminary setup lies in the fact that we now require the function $\phi_a^\star(\cdot)$ to be learned for each arm independently. The reward is generated as

$$r_{t,a_t} = \langle \boldsymbol{\theta}_{a_t}^\star, \boldsymbol{x}_{t,a_t} \rangle + \langle \boldsymbol{\beta}_{a_t}^\star, \boldsymbol{z}_{t,a_t} \rangle + \eta_t.$$

The following proposition shows our regret bound for this action-dependent context case. Its proof largely draws upon the proof idea of Theorem 1 and also relies on the generalized EPL Lemma 1.

**Proposition 2.** *The regret of poLinUCB in Algorithm 1 for action-dependent contexts is upper bounded by* $\widetilde{\mathcal{O}}\left(T^{1-\alpha} d_u^\alpha \sqrt{K} + d_u \sqrt{TK}\right)$ *with probability at least* $1-\delta$ *if* $T = \Omega(\log(1/\delta))$.

The main difference with the bound in Theorem 1 is the additional $\sqrt{K}$ appeared in the first term, which is caused by learning multiple $\phi_a^\star(\cdot)$ functions with $a \in \mathcal{A}$.

### 5.2. Generalization to Linear Stochastic Bandits

Another variant of linear bandits is the *linear stochastic bandits* setup (see, e.g., (Abbasi-Yadkori et al., 2011)). This model allows infinitely many arms, which consists of a decision set $D_t \subseteq \mathbb{R}^d$ at time $t$, and the learner picks an action $\boldsymbol{x}_t \in D_t$. This setup naturally generalizes to our problem with post-serving contexts. That is, at iteration $t$, the learner selects an arm $\boldsymbol{x}_t \in D_t$ first, receives reward $r_{t,\boldsymbol{x}_t}$, and then observe the post-serving feature $\boldsymbol{z}_t$ conditioned on $\boldsymbol{x}_t$. Similarly, we assume the existence of a mapping $\phi^\star(\boldsymbol{x}_t) = \mathbb{E}[\boldsymbol{z}_t | \boldsymbol{x}_z]$ that satisfies the Assumption 1. Consequently, the realized reward is generated as follows where $\boldsymbol{\theta}^*, \boldsymbol{\beta}^*$ are unknown parameters:

$$r_{t,\boldsymbol{x}_t} = \langle \boldsymbol{x}_t, \boldsymbol{\theta}^\star \rangle + \langle \boldsymbol{z}_t, \boldsymbol{\beta}^\star \rangle + \eta_t.$$

Therefore, the learner needs to estimate the linear parameters $\widehat{\boldsymbol{\theta}}$ and $\widehat{\boldsymbol{\beta}}$, as well as the function $\widehat{\phi}(\cdot)$. We obtain the following proposition for the this setup.

**Proposition 3.** *The regret of poLinUCB in Algorithm 1 for the above setting is upper bounded by* $\widetilde{\mathcal{O}}\left(T^{1-\alpha} d_u^\alpha + d_u \sqrt{T}\right)$ *with probability at least* $1 - \delta$ *if* $T = \Omega(\log(1/\delta))$.

### 5.3. Generalization to Linear Bandits with Feature Mappings

Finally, we briefly remark that while we have so far assumed that the arm parameters are directly linear in the context $\boldsymbol{x}_t, \boldsymbol{z}_t$, just like classic linear bandits our analysis can be easily generalized to accommodate feature mapping $\pi^x(\boldsymbol{x}_t)$ and $\pi^z(\boldsymbol{z}_t) = \pi^z(\phi(\boldsymbol{x}_t) + \varepsilon_t)$. Specifically, if the reward generation process is $r_a = \langle \boldsymbol{\theta}_a^\star, \pi^x(\boldsymbol{x}_t) \rangle + \langle \boldsymbol{\beta}_a^\star, \pi^z(\boldsymbol{z}_t) \rangle + \eta_t$ instead, then we can simply view $\tilde{\boldsymbol{x}}_t = \pi^x(\boldsymbol{x}_t)$ and $\tilde{\boldsymbol{z}}_t = \pi^z(\boldsymbol{z}_t)$ as the new features, with $\tilde{\phi}(\boldsymbol{x}_t) = \mathbb{E}_{\boldsymbol{\epsilon}_t}[\pi^z(\phi(\boldsymbol{x}_t) + \boldsymbol{\epsilon}_t)]$. By working with $\tilde{\boldsymbol{x}}_t, \tilde{\boldsymbol{z}}_t, \tilde{\phi}$, we shall obtain the same guarantees as Theorem 1.

## 6. Experiments

This section presents a comprehensive evaluation of our proposed poLinUCB algorithm on both synthetic and real-world data, demonstrating its effectiveness in incorporating follow-up information and outperforming the LinUCB($\widehat{\phi}$) variant.

### 6.1. Synthetic Data with Ground Truth Models

**Evaluation Setup.** We adopt three different synthetic environments that are representative of a range of mappings from the pre-serving context to the post-serving context: polynomial, periodicical and linear functions. The pre-serving contexts are sampled from a uniform noise in the range $[-10, 10]^{d_x}$, and Gaussian noise is employed for both the post-serving contexts and the rewards. In each environment, the dimensions of the pre-serving context ($d_x$) and the post-serving context ($d_z$) are of 100 and 5, respectively with 10 arms ($K$). The evaluation spans $T = 1000$ or $5000$ time steps, and each experiment is repeated with 10 different seeds. The cumulative regret for each policy in each environment is then calculated to provide a compararison.

**Results and Discussion.** Our experimental results, which are presented graphically in Figures 1, provide strong evidence of the superiority of our proposed poLinUCB algorithm. Across all setups, we observe that the LinUCB ($x$ and $z$) strategy, which has access to the post-serving context during arm selection, consistently delivers the best performance, thus serving as the upper bound for comparison. On the other hand, the Random policy, which does not exploit any environment information, performs the worst, serving as the lower bound. Our proposed poLinUCB (ours) outperforms all the other strategies, including the LinUCB ($\widehat{\phi}$) variant, in all three setups, showcasing its effectiveness in adaptively handling various mappings from the pre-serving context to the post-serving context. Importantly, poLinUCB delivers significantly superior performance to LinUCB ($x$ only), which operates solely based on the pre-serving context.

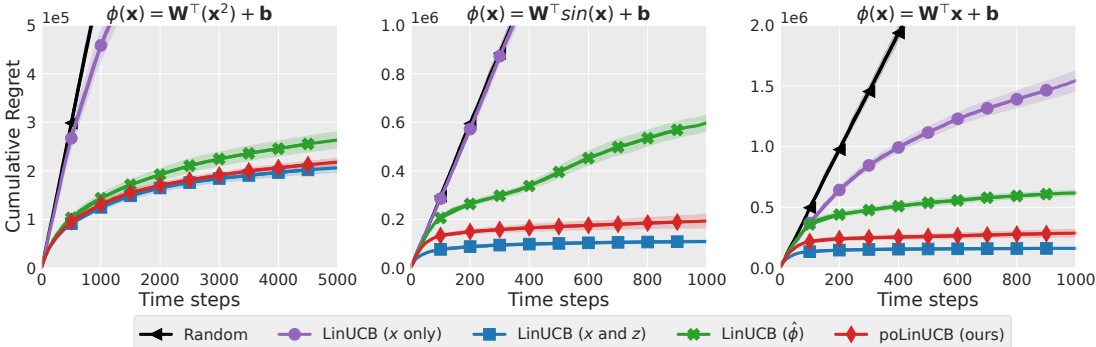

Figure 1: Cumulative Regret in three synthetic environments. Comparisons of different algorithms in terms of cumulative regret across the three synthetic environments. Our proposed poLinUCB (ours) consistently outperforms other strategies (except for LinUCB which has access to the post-serving context during arm selection), showcasing its effectiveness in utilizing post-serving contexts. The shaded area denotes the standard error computed using 10 different random seeds.

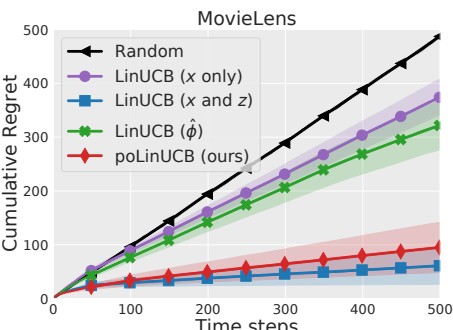

Figure 2: Results on MovieLens.

## 6.2. Real World Data without Ground Truth

**Evaluation Setup.** The evaluation was conducted on a real-world dataset, MovieLens (Harper and Konstan, 2015), where the task is to recommend movies (arms) to a incoming user (context). Following Yao et al. (2023), we first map both movies and users to 32-dimensional real vectors using a neural network trained for predicting the rating. Initially, $K = 5$ movies were randomly sampled to serve as our arms and were held fixed throughout the experiment. The user feature vectors were divided into two parts serving as the pre-serving context ($d_x = 25$) and the post-serving context ($d_z = 7$). We fit the function $\phi(\boldsymbol{x})$ using a two-layer neural network with 64 hidden units and ReLU activation. The network was trained using the Adam optimizer with a learning rate of 1e-3. At each iteration, we randomly sampled a user from the dataset and exposed only the pre-serving context $\boldsymbol{x}$ to our algorithm. The reward was computed as the dot product of the user's feature vector and the selected movie's feature vector and was revealed post the movie selection. The evaluation spanned $T = 500$ iterations and repeated with 10 seeds.

**Results and Discussion.** The experimental results, presented in Figure 2, demonstrate the effectiveness of our proposed algorithm. The overall pattern is similar to it ob-

served in our synthetic experiments. Our proposed policy consistently outperforms the other strategies (except for LinUCB with both pre-serving and post-serving features). Significantly, our algorithm yields superior performance compared to policies operating solely on the pre-serving context, thereby demonstrating its effectiveness in leveraging the post-serving information.

## 7. Conclusions and Limitations

In this work, we have introduced a novel contextual bandit framework that incorporates post-serving contexts, thereby widening the range of complex real-world challenges it can address. By leveraging historical data, our proposed algorithm, poLinUCB, estimates the functional mapping from pre-serving to post-serving contexts, leading to improved online learning efficiency. For the purpose of theoretical analysis, the elliptical potential lemma has been expanded to manage noise within post-serving contexts, a development which may have wider applicability beyond this particular framework. Extensive empirical tests on synthetic and real-world datasets have demonstrated the significant benefits of utilizing post-serving contexts and the superior performance of our algorithm compared to state-of-the-art approaches.

Our theoretical analysis hinges on the assumption that the function $\phi^\star(\cdot)$ is learnable, a condition that may not always be feasible. This especially applies to settings where the post-serving contexts may hold additional information that cannot be deduced from the pre-serving context, irrespective of the amount of data collected. In such scenarios, no function mapping from the pre-serving context to the post-serving context will satisfy the learnability assumption. Consequently, a linear regret is inevitable, and it will be proportional to the level of misspecification. However, from a practical point of view, our empirical findings from the real-world MovieLens dataset demonstrate that modeling the functional relationship between the pre-serving and post-serving contexts can significantly enhance recommendation.

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

# A. Extended Discussions

## A.1. Related Works

**Contextual bandits.** The literature on (generalized) linear (contextual) bandits is extensive, as evidenced by numerous studies (Abe et al., 2003; Auer, 2002; Dani et al., 2008; Rusmevichientong and Tsitsiklis, 2010; Lu et al., 2010; Filippi et al., 2010; Li et al., 2010; Chu et al., 2011; Abbasi-Yadkori et al., 2011; Li et al., 2017; Jun et al., 2017). These approaches predominantly employ upper confidence bounds as a means of balancing exploration and exploitation, leading to the attainment of minimax optimal regret bounds. The derivation of these regret bounds principally hinges on the utilization of confidence ellipsoids and the elliptical potential lemma. All these works assume that the contextual information governing the payoff is full observable. In contrast, our work focuses on scenarios where the context is not completely observable during arm selection, thereby presenting additional complexities in managing the partially available information.

**Contextual bandits with partial information.** Contextual bandits with partial information has been relatively limited in the literature. Initial progress in this area was made by Wang et al. (2016), who studied settings with hidden contexts. In their setup there is some context (the post-serving context in our model) that can never by observed by the learner, whereas in our setup the learner can observe post-serving context but only after pulling the arm. Under the assumption that if the parameter initialization is extremely close to the true optimal parameter, then they develop a sub-linear regret algorithm. Our algorithm does not need such strong assumption on parameter initialization and we also show that their approach may perform poorly in our setup. Subsequent research by Qi et al. (2018); Yang et al. (2020); Park and Faradonbeh (2021); Yang and Ren (2021); Zhu and Kveton (2022) investigated scenarios with noisy or unobservable contexts. In these studies, the learning algorithm was designed to predict context information online through context history analysis, or selectively request context data from an external expert. Our work, on the other hand, introduces a novel problem setting that separates contexts into pre-serving and post-serving categories, enabling the exploration of a wide range of problems with varying learnability. Additionally, we also need to employ new techniques for analyzing our problem to get a near-optimal regret bound.

**Generalizing the elliptical potential lemma (EPL).** The EPL, introduced by Lai and Wei (1982), serves as a critical component in quantifying the rate at which uncertainty decreases with the addition of more observations. Initially employed in the analysis of stochastic linear regression, the EPL has since been extensively utilized in stochastic linear bandit problems (Auer, 2002; Dani et al., 2008; Chu et al., 2011; Abbasi-Yadkori et al., 2011; Li et al., 2019; Zhou et al., 2020; Wang et al., 2022). Researchers have also proposed various generalizations of the EPL to accommodate diverse assumptions and problems. For example, (Carpentier et al., 2020) extended the EPL by allowing for the use of the $X_t^{-p}$-norm, as opposed to the traditional $X_t^{-1}$-norm. Meanwhile, (Hamidi and Bayati, 2022) investigated a generalized form of the $1 \wedge \|\varphi(\boldsymbol{x}_t)\|_{\boldsymbol{X}_{t-1}^{-1}}^2$ term, which was inspired by the pursuit of variance reduction in non-Gaussian linear regression models. However, existing (generalized) EPLs are inadequate for the analysis presented herein.

# B. Algorithm and Regret Analysis of LinUCB($\widehat{\phi}$)

We present the details of the algorithm described in Section 2.2 and the proof of the regret bound.

## B.1. Main Algorithm

**Parameter learning.** We consider solving the following regularized least squared problem for estimating $\{\widehat{\boldsymbol{\theta}}_{t,a}\}_{a \in \mathcal{A}}$ and $\{\widehat{\boldsymbol{\beta}}_{t,a}\}_{a \in \mathcal{A}}$ for each arm $a$:

$$\ell_t(\boldsymbol{\theta}_a, \boldsymbol{\beta}_a) = \sum_{s:a_s=a}^{t} \left( r_{s,a} - \boldsymbol{x}_t^\top \boldsymbol{\theta}_a - \widehat{\phi}_s(\boldsymbol{x}_s)^\top \boldsymbol{\beta}_a \right)^2 + \lambda \left( \|\boldsymbol{\theta}_a\|_2^2 + \|\boldsymbol{\beta}_a\|_2^2 \right), \tag{10}$$

where $\lambda \geq 0$ are penalty factors ensuring the uniqueness of minimizers $\widehat{\boldsymbol{\theta}}_{t,a}$ and $\widehat{\boldsymbol{\beta}}_{t,a}$.

In the same convention, we use $\boldsymbol{w}$ to denote $(\boldsymbol{\theta}, \boldsymbol{\beta})$, and $\boldsymbol{u}$ to denote $\left( \boldsymbol{x}, \widehat{\phi}(\boldsymbol{x}) \right)$. The closed-form solutions for $\widehat{\boldsymbol{\theta}}_{t,a}$ and $\widehat{\boldsymbol{\beta}}_{t,a}$ in this least squared problem then become:

$$\widehat{\boldsymbol{w}}_{t,a} := \begin{bmatrix} \widehat{\boldsymbol{\theta}}_{t,a} \\ \widehat{\boldsymbol{\beta}}_{t,a} \end{bmatrix} = \boldsymbol{A}_{t,a}^{-1} \boldsymbol{b}_{t,a},$$

where we reload the notations of $\boldsymbol{A}_{t,a}$ and $\boldsymbol{b}_{t,a}$,

$$\boldsymbol{A}_{t,a} = \lambda \boldsymbol{I} + \sum_{s:a_s=a}^{t} \boldsymbol{u}_s \boldsymbol{u}_s^\top \quad \text{and} \quad \boldsymbol{b}_{t,a} = \sum_{s:a_s=a}^{t} r_{s,a} \boldsymbol{u}_s. \tag{11}$$

**Confidence set construction.** At iteration $t$, we construct the confidence set for $\widehat{\phi}_t(\boldsymbol{x}_t)$ by

$$\mathcal{C}_t\left(\widehat{\phi}_t, \boldsymbol{x}_t\right) := \left\{ \boldsymbol{z} \in \mathbb{R}^d : \left\| \widehat{\phi}_t(\boldsymbol{x}_t) - \phi^\star(\boldsymbol{x}_t) \right\|_2 \le e_t^\delta \right\}. \tag{12}$$

The construction of the confidence set for $\widehat{\boldsymbol{w}}_{t,a}$ will be different, as we are using the predicted value $\widehat{\phi}_t(\cdot)$ for linear regression. Consider the following,

$$\boldsymbol{A}_{t,a} \left( \begin{bmatrix} \widehat{\boldsymbol{\theta}}_{t,a} \\ \widehat{\boldsymbol{\beta}}_{t,a} \end{bmatrix} - \begin{bmatrix} \boldsymbol{\theta}_a^\star \\ \boldsymbol{\beta}_a^\star \end{bmatrix} \right) = \underbrace{\sum_{s:a_s=a}^{t} \left( \boldsymbol{\epsilon}_s^\top \boldsymbol{\beta}_a^\star \begin{bmatrix} \boldsymbol{x}_s \\ \widehat{\phi}_s(\boldsymbol{x}_s) \end{bmatrix} \right)}_{\textcircled{1}} + \underbrace{\sum_{s:a_s=a}^{t} \left( \left( \phi^\star(\boldsymbol{x}_s) - \widehat{\phi}_s(\boldsymbol{x}_s) \right)^\top \boldsymbol{\beta}_a^\star \begin{bmatrix} \boldsymbol{x}_s \\ \widehat{\phi}_s(\boldsymbol{x}_s) \end{bmatrix} \right)}_{\textcircled{2}}$$

$$+ \underbrace{\sum_{s:a_s=a}^{t} \eta_s \begin{bmatrix} \boldsymbol{x}_s \\ \widehat{\phi}_s(\boldsymbol{x}_s) \end{bmatrix}}_{\textcircled{3}} - \underbrace{\lambda \begin{bmatrix} \boldsymbol{\theta}_a^\star \\ \boldsymbol{\beta}_a^\star \end{bmatrix}}_{\textcircled{4}}$$

Therefore, the confidence set will be enlarged due to the error introduced by $\widehat{\phi}_t(\cdot)$. In the below, we derive the confidence set. To build the confidence set, we need to bound

$$\|\widehat{\boldsymbol{w}}_{t,a} - \boldsymbol{w}_{t,a}^\star\|_{\boldsymbol{A}_{t,a}} = \left\| \begin{bmatrix} \widehat{\boldsymbol{\theta}}_{t,a} \\ \widehat{\boldsymbol{\beta}}_{t,a} \end{bmatrix} - \begin{bmatrix} \boldsymbol{\theta}_a^\star \\ \boldsymbol{\beta}_a^\star \end{bmatrix} \right\|_{\boldsymbol{A}_{t,a}} = \left\| \textcircled{1} + \textcircled{2} + \textcircled{3} + \textcircled{4} \right\|_{\boldsymbol{A}_{t,a}^{-1}}$$

Since both $\{\boldsymbol{\epsilon}_s\}_{s=1}^t$ and $\{\eta_s\}_{s=1}^t$ are i.i.d sub-Gaussian random variables, respectively, we can use the self-normalized inequality to bound the corresponding terms, i.e., the followings hold with probability at least $1 - 2\delta$,

$$\|\widehat{\boldsymbol{w}}_{t,a} - \boldsymbol{w}_{t,a}^\star\|_{\boldsymbol{A}_{t,a}} \le \sqrt{2(L_\epsilon^2 + R_\eta^2) \log\left( \frac{\det(\boldsymbol{A}_{t,a})^{1/2} \det(\lambda \boldsymbol{I})^{-1/2}}{\delta/2} \right)} + \frac{L_u}{\sqrt{\lambda}} \left( \sum_{s=1}^t e_s^{\delta/t} \right) + 2\sqrt{\lambda}$$

$$\le \sqrt{2(L_\epsilon^2 + R_\eta^2) \log\left( \frac{1 + n_t(a)L_u^2/\lambda}{\delta/2} \right)} + \frac{L_u}{\sqrt{\lambda}} \left( \sum_{s=1}^t e_s^{\delta/t} \right) + 2\sqrt{\lambda}$$

Therefore, the confidence set is

$$\mathcal{C}_{t,a}(\widehat{\boldsymbol{w}}_{t,a}) = \left\{ \boldsymbol{w} \in \mathbb{R}^{d_u} : \|\boldsymbol{w} - \widehat{\boldsymbol{w}}_{t,a}\|_{\boldsymbol{A}_{t,a}} \le \zeta_{t,a} \right\}, \tag{13}$$

where $\zeta_{t,a} = \sqrt{2(L_\epsilon^2 + R_\eta^2) \log\left((1 + n_t(a)L_u^2/\lambda)/(\delta/2)\right)} + L_u \left( \sum_{s=1}^t e_s^{\delta/t} \right) / \sqrt{\lambda} + 2\sqrt{\lambda}$. In comparison to the original confidence set, there is one additional term due to the generalization error introduced from $\widehat{\phi}_s(\cdot)$. In the next section, we

**Algorithm 2** LinUCB-$(\widehat{\phi})$ (*Linear UCB adapted from* Wang et al. (2016) *with post-serving contexts; The differences with Algorithm* 1 *are highlighted in* blue *color.*)

1: **for** $t = 0, 1, \ldots, T$ **do**
2:    Receive the *pre-serving context* $\boldsymbol{x}_t$
3:    Compute the optimistic parameters by maximizing the UCB objective

$$\left( a_t, \widetilde{\phi}_t(\boldsymbol{x}_t), \widetilde{\boldsymbol{w}}_t \right) = \underset{(a, \phi, \boldsymbol{w}_a) \in [K] \times \mathcal{C}_{t-1}\left(\widehat{\phi}_{t-1}, \boldsymbol{x}_t\right) \times \mathcal{C}_{t-1}(\widehat{\boldsymbol{w}}_{t-1,a})}{\arg\max} \begin{bmatrix} \boldsymbol{x}_t \\ \phi(\boldsymbol{x}_t) \end{bmatrix}^\top \boldsymbol{w}_a .$$

4:    Play the arm $a_t$ and receive the realized *post-serving context* as $\boldsymbol{z}_t$ and the real-valued reward

$$r_{t,a_t} = \begin{bmatrix} \boldsymbol{x}_t \\ \boldsymbol{z}_t \end{bmatrix}^\top \boldsymbol{w}_{a_t}^\star + \eta_t .$$

5:    Compute the estimated post-serving context generating function $\widehat{\phi}_t(\cdot)$ using ERM.
6:    Compute $\widehat{\boldsymbol{w}}_{t,a}$ by solving Equation 10 for each $a$.
7:    Update confidence sets $\mathcal{C}_t(\widehat{\boldsymbol{w}}_{t,a})$ and $\mathcal{C}_t(\widehat{\phi}_t, \boldsymbol{x}_t)$ for each $a$ based on Equations 12 and 5.
8: **end for**

will provide a regret analysis, which following from the proof of LinUCB (Li et al., 2010). We simply have the following

$$R_T = \sum_{t=1}^{T} \left( r_{t,a_t^\star} - r_{t,a_t} \right) = \sum_{t=1}^{T} \Delta_t \leq \sqrt{T \sum_{t=1}^{T} \Delta_t^2}$$

$$\leq \sqrt{T \sum_{t=1}^{T} \left( \left\| \widetilde{\phi}_t(\boldsymbol{x}_t) - \phi^\star(\boldsymbol{x}_t) \right\| \left\| \widetilde{\boldsymbol{\beta}}_{a_t} \right\| + \left\| \begin{bmatrix} \boldsymbol{x}_t \\ \phi^\star(\boldsymbol{x}_t) \end{bmatrix} \right\|_{\boldsymbol{A}_{t-1,a_t}^{-1}} \left\| \begin{bmatrix} \widetilde{\boldsymbol{\theta}}_{a_t} - \boldsymbol{\theta}_{a_t} \\ \widetilde{\boldsymbol{\beta}}_{a_t} - \boldsymbol{\beta}_{a_t} \end{bmatrix} \right\|_{\boldsymbol{A}_{t-1,a_t}} \right)^2}$$

$$\leq \sqrt{T \left( \sum_{t=1}^{T} 2 \left\| \widetilde{\phi}_t(\boldsymbol{x}_t) - \phi^\star(\boldsymbol{x}_t) \right\|^2 \left\| \widetilde{\boldsymbol{\beta}}_{a_t} \right\|^2 + 2\zeta_{T,a}^2 \left( 1 \wedge \left\| \begin{bmatrix} \boldsymbol{x}_t \\ \phi^\star(\boldsymbol{x}_t) \end{bmatrix} \right\|_{\boldsymbol{A}_{t-1,a_t}^{-1}}^2 \right) \right)}$$

$$\leq \sqrt{T \left( \sum_{t=1}^{T} 2 \left\| \widetilde{\phi}_t(\boldsymbol{x}_t) - \phi^\star(\boldsymbol{x}_t) \right\|^2 \left\| \widetilde{\boldsymbol{\beta}}_{a_t} \right\|^2 + 2\zeta_{T,a}^2 \left( 1 \wedge \left\| \begin{bmatrix} \boldsymbol{x}_t \\ \phi^\star(\boldsymbol{x}_t) \end{bmatrix} \right\|_{\boldsymbol{A}_{t-1,a_t}^{-1}}^2 \right) \right)}$$

$$\leq \sqrt{T \cdot \left( 8C_0 T^{1-2\alpha} \log^{2\alpha} \left( \frac{\det \boldsymbol{X}_t}{\det \boldsymbol{X}_0} \right) \log^2 \left( \frac{T}{\delta} \right) + 2K \zeta_T^2 d_u \log \left( 1 + \frac{TL_u^2}{\lambda d_u} \right) \right)}$$

In the next, we expand the term $\zeta_T^2$,

$$\zeta_T^2 \leq \left( \sqrt{2(L_\epsilon^2 + R_\eta^2) \log \left( \frac{1 + TL_u^2/\lambda}{\delta/2} \right)} + \frac{L_u \left( \sum_{s=1}^{T} e_s^{\delta/T} \right)}{\sqrt{\lambda}} + 2\sqrt{\lambda} \right)^2$$

$$\leq 6(L_\epsilon^2 + R_\eta^2) \log \left( \frac{1 + TL_u^2/\lambda}{\delta/2} \right) + \frac{3L_u^2}{\lambda} \left( \sum_{s=1}^{T} e_s^{\delta/T} \right)^2 + 12\lambda$$

In the next, we bound the second term in the above equation under the learnability assumption 1,

$$\left( \sum_{s=1}^{T} e_s^{\delta/T} \right)^2 \leq 16T^{2-2\alpha} \log^{2\alpha} \left( \frac{\det \boldsymbol{X}_T}{\det \boldsymbol{X}_0} \right) \log^2 \left( \frac{T}{\delta} \right) .$$

Therefore, naively choosing the value of $\lambda$ will lead to a linear regret due to the term $T^{3/2-\alpha}$ in the equation. To minimize the upper bound, we can choose the value of $\lambda$ to be

$$\lambda = 2L_u T^{1-\alpha} \log^\alpha \left( \frac{\det \boldsymbol{X}_T}{\det \boldsymbol{X}_0} \right) \log \left( \frac{T}{\delta} \right).$$

Then, we can bound $\zeta_T^2$ by

$$\zeta_T^2 \le 6(L_\epsilon^2 + R_\eta^2) \log \left( \frac{1 + TL_u^2/\lambda}{\delta/2} \right) + 48 L_u T^{1-\alpha} \log^\alpha \left( \frac{\det \boldsymbol{X}_T}{\det \boldsymbol{X}_0} \right) \log \left( \frac{T}{\delta} \right).$$

By plugging it in and following the simplication as in the proof of Theorem 1, we can get the regret is upper bounded by

$$\widetilde{\mathcal{O}} \left( T^{1-\alpha} d_u^\alpha + T^{1-\alpha/2} \sqrt{K d_u^{1+\alpha}} \right).$$

The above result is summarized as the following proposition.

**Proposition 1** (Regret of LinUCB-$(\widehat{\phi})$)**.** *The regret of LinUCB-$(\widehat{\phi})$ in Algorithm 2 is upper bounded by* $\widetilde{\mathcal{O}} \left( T^{1-\alpha} d_u^\alpha + T^{1-\alpha/2} \sqrt{K d_u^{1+\alpha}} \right)$ *with probability at least $1 - \delta$, by carefully setting the regularization coefficient* $\lambda = \Theta(L_u d_u^\alpha T^{1-\alpha} \log (T/\delta))$ *in Equation 1.*

## C. Missing Proofs

### C.1. Missing Proofs in the Generalized Elliptical Potential Lemma

**Lemma 1** (Generalized Elliptical Potential Lemma)**.** *Suppose (1) $\boldsymbol{X}_0 \in \mathbb{R}^{d \times d}$ is any positive definite matrix; (2) $\boldsymbol{x}_1, \ldots, \boldsymbol{x}_T \in \mathbb{R}^d$ is a sequence of vectors with bounded $l_2$ norm $\max_t \|\boldsymbol{x}_t\| \le L_x$; (3) $\boldsymbol{\epsilon}_1, \ldots, \boldsymbol{\epsilon}_T \in \mathbb{R}^d$ is a sequence of independent (not necessarily identical) bounded zero-mean noises satisfying $\max_t \|\boldsymbol{\epsilon}_t\| \le L_\epsilon$ and $\mathbb{E}[\boldsymbol{\epsilon}_t \boldsymbol{\epsilon}_t^\top] \succcurlyeq \sigma_\epsilon^2 \boldsymbol{I}$ for any $t$; and (4) $\widetilde{\boldsymbol{X}}_t$ is defined as follows:*

$$\widetilde{\boldsymbol{X}}_t = \boldsymbol{X}_0 + \sum_{s=1}^t (\boldsymbol{x}_s + \boldsymbol{\epsilon}_s)(\boldsymbol{x}_s + \boldsymbol{\epsilon}_s)^\top \in \mathbb{R}^{d \times d}.$$

*Then, for any $p \in [0, 1]$, the following inequality holds with probability at least $1 - \delta$,*

$$\sum_{t=1}^T \left( 1 \wedge \|\boldsymbol{x}_t\|_{\widetilde{\boldsymbol{X}}_{t-1}^{-1}}^2 \right)^p \le 2^p T^{1-p} \log^p \left( \frac{\det \boldsymbol{X}_T}{\det \boldsymbol{X}_0} \right)$$
$$+ \frac{8 L_\epsilon^2 (L_\epsilon + L_x)^2}{\sigma_\epsilon^4} \log \left( \frac{32 d L_\epsilon^2 (L_\epsilon + L_x)^2}{\delta \sigma_\epsilon^4} \right) \tag{2}$$

*Proof.* Our proof follows the high level idea for proving the original EPL. However, to accommodate the noises in the data matrix, we have to introduce new matrix concentration tools to the original (primarily algebraic) proof, and also identify the right conditions for the argument to go through. A key lemma to our proof is the following high probability bound regarding the noisy data matrix:

**Lemma 2.** *Let $\boldsymbol{x}_1, ..., \boldsymbol{x}_T \in \mathbb{R}^d$ be a fixed sequence of vectors, and $\boldsymbol{\epsilon}_1, ..., \boldsymbol{\epsilon}_T \in \mathbb{R}^d$ are independent random variables satisfying $\max_t \|\boldsymbol{x}_t\|_2 \le L_x$, $\max_t \|\boldsymbol{\epsilon}_t\|_2 \le L_\epsilon$, and $\mathbb{E}[\boldsymbol{\epsilon}_t \boldsymbol{\epsilon}_t^\top] \succcurlyeq \sigma_\epsilon^2 \boldsymbol{I}$. Then the following hold with probability at least $1 - 2d \exp \left( \frac{-T \sigma_\epsilon^4}{8 L_\epsilon^2 (L_\epsilon + L_x)^2} \right)$,*

$$\sum_{t=1}^T (\boldsymbol{x}_t + \boldsymbol{\epsilon}_t)(\boldsymbol{x}_t + \boldsymbol{\epsilon}_t)^\top \succcurlyeq \sum_{t=1}^T \boldsymbol{x}_t \boldsymbol{x}_t^\top.$$

The Proof of Lemma 2 employs the Bernstein's Inequality for matrices (Tropp et al., 2015), which is technical; for ease of presentation, we defer its proof of Appendix C.1. By Lemma 2, we have that, for every $t \in [T]$, the following inequality holds with probability at least $1 - 2d \exp\left(\frac{-t\sigma_\epsilon^4}{8L_\epsilon^2(L_\epsilon + L_x)^2}\right)$:

$$\widetilde{\boldsymbol{X}}_t = \boldsymbol{X}_0 + \sum_{s=1}^{t}(\boldsymbol{x}_s + \boldsymbol{\epsilon}_s)(\boldsymbol{x}_s + \boldsymbol{\epsilon}_s)^\top \succcurlyeq \boldsymbol{X}_0 + \sum_{s=1}^{t}\boldsymbol{x}_s\boldsymbol{x}_s^\top := \boldsymbol{X}_t,$$

under which we have

$$\|\boldsymbol{x}_{t+1}\|_{\widetilde{\boldsymbol{X}}_t^{-1}}^2 \le \|\boldsymbol{x}_{t+1}\|_{\boldsymbol{X}_t^{-1}}^2.$$

To prove our Lemma 1, we need to apply union bound to guarantee the above hold simultaneously for every $t \ge 1$ with high probability. Unfortunately, this turns out to not be true because when $t$ is very small (e.g., $t = 1$), the above inequality cannot hold with high probability. Therefore, to obtain high-probability guarantee by the union bound, we will have to exclude these small $t$'s and apply the union bound for only the events from some $t' \in [T]$, as follows

$$\mathbb{P}\left[\forall t \in [t', T], \ \|\boldsymbol{x}_t\|_{\widetilde{\boldsymbol{X}}_{t-1}^{-1}}^2 \le \|\boldsymbol{x}_t\|_{\boldsymbol{X}_{t-1}^{-1}}^2\right]$$

$$\ge 1 - \sum_{t=t'-1}^{T-1} 2d \exp\left(\frac{-t\sigma_\epsilon^4}{(8L_\epsilon^2(L_\epsilon + L_x)^2)}\right)$$

$$\ge 1 - \sum_{t=t'-1}^{\infty} 2d \exp\left(\frac{-t\sigma_\epsilon^4}{(8L_\epsilon^2(L_\epsilon + L_x)^2)}\right)$$

$$= 1 - 2d\left(\frac{\exp\left(-(t'-1)\sigma_\epsilon^4/(8L_\epsilon^2(L_\epsilon + L_x)^2)\right)}{1 - \exp\left(-\sigma_\epsilon^4/(8L_\epsilon^2(L_\epsilon + L_x)^2)\right)}\right).$$

$$\ge 1 - 2d \times \exp\left(-(t'-1)\sigma_\epsilon^4/(8L_\epsilon^2(L_\epsilon + L_x)^2)\right) \times \frac{16L_\epsilon^2(L_\epsilon + L_x)^2}{\sigma_\epsilon^4},$$

where the last inequality uses the fact that $\sigma_\epsilon^4/(L_\epsilon^2(L_\epsilon + L_x)^2) \le (\sigma_\epsilon/L_\epsilon)^4 \le 1$ and $1 - e^{-x} \ge x/2$ for any $x \in [0, 1]$. By solving the following inequality,

$$\exp\left(-(t'-1)\sigma_\epsilon^4/(8L_\epsilon^2(L_\epsilon + L_x)^2)\right) \times \frac{16L_\epsilon^2(L_\epsilon + L_x)^2}{\sigma_\epsilon^4} \le \frac{\delta}{2d},$$

we have,

$$t' \ge 1 + \frac{8L_\epsilon^2(L_\epsilon + L_x)^2}{\sigma_\epsilon^4} \log\left(\frac{32dL_\epsilon^2(L_\epsilon + L_x)^2}{\delta\sigma_\epsilon^4}\right)$$

Let $T_0$ denote the ceiling of the right-hand-side of the above term. Therefore, we have the following hold with high probability at least $1 - \delta$:

$$\sum_{t=1}^{T}\left(1 \wedge \|\boldsymbol{x}_t\|_{\widetilde{\boldsymbol{X}}_{t-1}^{-1}}^2\right)^p \le (T_0 - 1) + \sum_{t=T_0}^{T}\left(1 \wedge \|\boldsymbol{x}_t\|_{\widetilde{\boldsymbol{X}}_{t-1}^{-1}}^2\right)^p$$

$$\le (T_0 - 1) + \sum_{t=T_0}^{T}\left(1 \wedge \|\boldsymbol{x}_t\|_{\boldsymbol{X}_{t-1}^{-1}}^2\right)^p$$

$$\le (T_0 - 1) + \sum_{t=T_0}^{T}\left(1 \wedge \|\boldsymbol{x}_t\|_{\boldsymbol{X}_{t-1}^{-1}}^2\right)^p$$

In the next, we are going to bound the second term in the above equation, whose proof can be adapted from the proof of the original elliptical potential lemma. Using the fact that for any $z \in [0, +\infty]$, $z \wedge 1 \le 2\ln(1 + z)$, we get

$$\sum_{t=1}^{T} 1 \wedge \left(\|\boldsymbol{x}_t\|_{\boldsymbol{X}_{t-1}^{-1}}^2\right)^p \le \sum_{t=1}^{T}\left(2\log\left(1 + \|\boldsymbol{x}_t\|_{\boldsymbol{X}_{t-1}^{-1}}^2\right)\right)^p.$$

Additionally, by definition, we have

$$\boldsymbol{X}_t = \boldsymbol{X}_{t-1} + \boldsymbol{x}_t \boldsymbol{x}_t^\top = \boldsymbol{X}_{t-1}^{1/2} \left( \boldsymbol{I} + \boldsymbol{X}_{t-1}^{-1/2} \boldsymbol{x}_t \boldsymbol{x}_t^\top \boldsymbol{X}_{t-1}^{-1/2} \right) \boldsymbol{X}_{t-1}^{1/2}.$$

This implies the following relationship between the determinant,

$$\det \boldsymbol{X}_t = \det \left( \boldsymbol{X}_{t-1} \right) \det \left( \boldsymbol{I} + \boldsymbol{X}_{t-1}^{-1/2} \boldsymbol{x}_t \boldsymbol{x}_t^\top \boldsymbol{X}_{t-1}^{-1/2} \right).$$

Since the only eigenvalues of a matrix of the form $\boldsymbol{I} + \boldsymbol{y}\boldsymbol{y}^\top$ are $1 + \|\boldsymbol{y}\|_2$ and $1$, we have

$$\log \left( 1 + \|\boldsymbol{x}_t\|_{\boldsymbol{X}_{t-1}^{-1}}^2 \right) = \log \det \boldsymbol{X}_t - \log \det \boldsymbol{X}_{t-1}.$$

By taking the power $p$ for both sides and taking the sum, we have

$$\sum_{t=1}^{T} \left( \log \left( 1 + \|\boldsymbol{x}_t\|_{\boldsymbol{X}_{t-1}^{-1}}^2 \right) \right)^p = \sum_{t=1}^{T} \left( \log \det \boldsymbol{X}_t - \log \det \boldsymbol{X}_{t-1} \right)^p.$$

Since $p \in [0,1]$, the function $g(x) = x^p$ is a concave function. Thus, we have

$$\frac{1}{T} \sum_{t=1}^{T} \left( \log \det \boldsymbol{X}_t - \log \det \boldsymbol{X}_{t-1} \right)^p \le \left( \frac{1}{T} \sum_{t=1}^{T} \log \det \boldsymbol{X}_t - \log \det \boldsymbol{X}_{t-1} \right)^p = \frac{1}{T^p} \log^p \left( \frac{\det \boldsymbol{X}_T}{\det \boldsymbol{X}_0} \right).$$

Therefore, we can conclude that

$$\sum_{t=1}^{T} 1 \wedge \left( \|\boldsymbol{x}_t\|_{\boldsymbol{X}_{t-1}^{-1}}^2 \right)^p \le 2^p T^{1-p} \log^p \left( \frac{\det \boldsymbol{X}_T}{\det \boldsymbol{X}_0} \right).$$

By combining the above results, we have the following hold with probability at least $1 - \delta$:

$$\sum_{t=1}^{T} \left( 1 \wedge \|\boldsymbol{x}_t\|_{\widetilde{\boldsymbol{X}}_{t-1}^{-1}}^2 \right)^p \le T_0 - 1 + \left( \sum_{t=1}^{T} 1 \wedge \|\boldsymbol{x}_t\|_{\boldsymbol{X}_{t-1}^{-1}}^2 \right)^p \le T_0 - 1 + 2^p T^{1-p} \log^p \left( \frac{\det \boldsymbol{X}_T}{\det \boldsymbol{X}_0} \right).$$

Invoking

$$T_0 - 1 \le \frac{8 L_\epsilon^2 (L_\epsilon + L_x)^2}{\sigma_\epsilon^4} \log \left( \frac{32 d L_\epsilon^2 (L_\epsilon + L_x)^2}{\delta \sigma_\epsilon^4} \right),$$

we obtained the desired inequality with probability at least $1 - \delta$:

$$\sum_{t=1}^{T} \left( 1 \wedge \|\boldsymbol{x}_t\|_{\widetilde{\boldsymbol{X}}_{t-1}^{-1}}^2 \right)^p \le 2^p T^{1-p} \log^p \left( \frac{\det \boldsymbol{X}_T}{\det \boldsymbol{X}_0} \right) + \frac{8 L_\epsilon^2 (L_\epsilon + L_x)^2}{\sigma_\epsilon^4} \log \left( \frac{32 d L_\epsilon^2 (L_\epsilon + L_x)^2}{\delta \sigma_\epsilon^4} \right).$$

$\square$

**Lemma 2.** *Let $\boldsymbol{x}_1, ..., \boldsymbol{x}_T \in \mathbb{R}^d$ be a fixed sequence of vectors, and $\boldsymbol{\epsilon}_1, ..., \boldsymbol{\epsilon}_T \in \mathbb{R}^d$ are independent random variables satisfying $\max_t \|\boldsymbol{x}_t\|_2 \le L_x$, $\max_t \|\boldsymbol{\epsilon}_t\|_2 \le L_\epsilon$, and $\mathbb{E}[\boldsymbol{\epsilon}_t \boldsymbol{\epsilon}_t^\top] \succcurlyeq \sigma_\epsilon^2 \boldsymbol{I}$. Then the following hold with probability at least $1 - 2d \exp \left( \frac{-T \sigma_\epsilon^4}{8 L_\epsilon^2 (L_\epsilon + L_x)^2} \right)$,*

$$\sum_{t=1}^{T} (\boldsymbol{x}_t + \boldsymbol{\epsilon}_t)(\boldsymbol{x}_t + \boldsymbol{\epsilon}_t)^\top \succcurlyeq \sum_{t=1}^{T} \boldsymbol{x}_t \boldsymbol{x}_t^\top.$$

*Proof.* We will analyze the term-wise difference, denoted as

$$S_t := (x_t + \epsilon_t)(x_t + \epsilon_t)^\top - x_t x_t^\top = \epsilon_t x_t^\top + x_t \epsilon_t^\top + \epsilon_t \epsilon_t^\top \tag{14}$$

Moreover, since $\mathbb{E}[\epsilon_t] = 0$ for any $t \in [T]$, the expectation of $S_t$ (over randomness of noise) can be lower bounded as

$$\mathbb{E}[S_t] = \mathbb{E}[\epsilon_t]x_t^\top + x_t\mathbb{E}[\epsilon_t^\top] + \mathbb{E}[\epsilon_t\epsilon_t^\top] = \mathbb{E}[\epsilon_t\epsilon_t^\top] \succcurlyeq \sigma_\epsilon^2 I.$$

Since $\|x_t\|_2 \leq L_x$ and $\|\epsilon_t\|_2 \leq L_\epsilon$, we know that $S_t \preccurlyeq (2L_\epsilon L_x + L_\epsilon^2)I$ with probability 1. Thus $S_t$ is uniformly upper bounded under the spectral-norm denoted by $\|\cdot\|$, or formally

$$\|S_t\| \leq 2L_\epsilon L_x + L_\epsilon^2.$$

Consider the "centered" matrix sum $Z_T = \sum_{t=1}^T \left[ S_t - \mathbb{E}[\epsilon_t\epsilon_t^\top] \right]$, with mean 0. Since the spectral-norm of $S_t$ is upper bounded by $2L_\epsilon L_x + L_\epsilon^2$, its variance $\mathbb{V}(S_t) = \left\| \mathbb{E}\left[ S_t S_t^\top \right] \right\|_2$ is upper bounded by $(2L_\epsilon L_x + L_\epsilon^2)^2$. Thus, the variance of $Z_T$ equals the variance of sum $\sum_{t=1}^T S_t$, which is then upper bounded by $T(2L_\epsilon L_x + L_\epsilon^2)^2$. By the Bernstein's Inequality for random matrices (Tropp et al., 2015), we have the following high probability upper bound for the spectral norm $\|\cdot\|$ of $Z_T$:

$$\mathbb{P}\left[ \|Z_T\| \geq \iota \right] \leq 2d \exp\left( \frac{-\iota^2/2}{\mathbb{V}(Z_T) + (2L_\epsilon L_x + L_\epsilon^2)\iota/3} \right). \tag{15}$$

Since $Z_T$ is a symmetric matrix, its spectral norm upper bounds the absolute value of any eigenvalue. Thus we can lower bound the smallest eigenvalues as follows:

$$\mathbb{P}\left[ \|Z_T\| \geq \iota \right] \geq \mathbb{P}\left[ \lambda_{\min}\left( \sum_{t=1}^T S_t - \mathbb{E}[\epsilon_t\epsilon_t^\top] \right) \leq -\iota \right]$$

$$\geq \mathbb{P}\left[ \lambda_{\min}\left( \sum_{t=1}^T S_t - \sigma_\epsilon^2 I \right) \leq -\iota \right]$$

$$= \mathbb{P}\left[ \lambda_{\min}\left( \sum_{t=1}^T S_t \right) \leq T\sigma_\epsilon^2 - \iota \right].$$

where the second inequality is due to two facts: (1) $A \succcurlyeq B$ implies $\lambda_{\min}(A) \geq \lambda_{\min}(B)$ where $A = \left( \sum_{t=1}^T S_t - \sigma_\epsilon^2 I \right)$ and $B = \left( \sum_{t=1}^T S_t - \mathbb{E}[\epsilon_t\epsilon_t^\top] \right)$; and (2) the event $\lambda_{\min}(A) \leq -\iota$ thus is included within the event $\lambda_{\min}(B) \leq -\iota$. Consequently, we have

$$\mathbb{P}\left[ \lambda_{\min}\left( \sum_{t=1}^T S_t \right) \leq T\sigma_\epsilon^2 - \iota \right] \leq 2d\exp\left( \frac{-\iota^2/2}{\mathbb{V}(Z_T) + (2L_\epsilon L_x + L_\epsilon^2)\iota/3} \right),$$

or equivalently,

$$\mathbb{P}\left[ \lambda_{\min}\left( \sum_{t=1}^T S_t \right) \geq T\sigma_\epsilon^2 - \iota \right] \geq 1 - 2d\exp\left( \frac{-\iota^2/2}{\mathbb{V}(Z_T) + (2L_\epsilon L_x + L_\epsilon^2)\iota/3} \right),$$

By choosing the value of $\iota = T\sigma_\epsilon^2$, we get

$$\mathbb{P}\left[ \lambda_{\min}\left( \sum_{t=1}^T S_t \right) \geq 0 \right]$$

$$\geq 1 - 2d\exp\left( \frac{-T^2\sigma_\epsilon^4/2}{\mathbb{V}(Z_T) + (2L_\epsilon L_x + L_\epsilon^2)T\sigma_\epsilon^2/3} \right)$$

$$\geq 1 - 2d\exp\left( \frac{-T^2\sigma_\epsilon^4/2}{T(2L_\epsilon L_x + L_\epsilon^2)^2 + (2L_\epsilon L_x + L_\epsilon^2)T\sigma_\epsilon^2/3} \right).$$

Using the fact that $\sigma_\epsilon \leq L_\epsilon$, we can further simplify the above equation by

$$\mathbb{P}\left[\lambda_{\min}\left(\sum_{t=1}^{T}\boldsymbol{S}_t\right) \geq 0\right] \geq 1 - 2d\exp\left(\frac{-T\sigma_\epsilon^4}{8L_\epsilon^2(L_\epsilon + L_x)^2}\right).$$

$\square$

### C.2. Missing Proofs in the Regret Analysis

**Theorem 1** (Regret of poLinUCB). *The regret of poLinUCB in Algorithm 1 is upper bounded by* $\widetilde{\mathcal{O}}\left(T^{1-\alpha}d_u^\alpha + d_u\sqrt{TK}\right)$ *with probability at least* $1 - \delta$, *if* $T = \Omega(\log(1/\delta))$.

*Proof.* In the next, we prove the regret bound. For each time step $t$, the immediate regret is

$$
\begin{aligned}
\Delta_t &= r_{t,a_t^\star} - r_{t,a_t}\\
&= \left\langle\begin{bmatrix}\boldsymbol{x}_t\\\phi^\star(\boldsymbol{x}_t)\end{bmatrix}, \begin{bmatrix}\boldsymbol{\theta}_{a_t^\star} - \boldsymbol{\theta}_{a_t}^\star\\\boldsymbol{\beta}_{a_t^\star} - \boldsymbol{\beta}_{a_t}^\star\end{bmatrix}\right\rangle\\
&\overset{(a)}{\leq} \left\langle\begin{bmatrix}\boldsymbol{x}_t\\\widetilde{\phi}_t(\boldsymbol{x}_t)\end{bmatrix}, \begin{bmatrix}\widetilde{\boldsymbol{\theta}}_{a_t} - \boldsymbol{\theta}_{a_t}^\star\\\widetilde{\boldsymbol{\beta}}_{a_t} - \boldsymbol{\beta}_{a_t}^\star\end{bmatrix}\right\rangle + \left\langle\widetilde{\phi}_t(\boldsymbol{x}_t) - \phi^\star(\boldsymbol{x}_t), \boldsymbol{\beta}_{a_t}^\star\right\rangle\\
&= \left\langle\begin{bmatrix}\boldsymbol{0}\\\widetilde{\phi}_t(\boldsymbol{x}_t) - \phi^\star(\boldsymbol{x}_t)\end{bmatrix} + \begin{bmatrix}\boldsymbol{x}_t\\\phi^\star(\boldsymbol{x}_t)\end{bmatrix}, \begin{bmatrix}\widetilde{\boldsymbol{\theta}}_{a_t} - \boldsymbol{\theta}_{a_t}^\star\\\widetilde{\boldsymbol{\beta}}_{a_t} - \boldsymbol{\beta}_{a_t}^\star\end{bmatrix}\right\rangle + \left\langle\widetilde{\phi}_t(\boldsymbol{x}_t) - \phi^\star(\boldsymbol{x}_t), \boldsymbol{\beta}_{a_t}^\star\right\rangle\\
&= \left\langle\widetilde{\phi}_t(\boldsymbol{x}_t) - \phi^\star(\boldsymbol{x}_t), \widetilde{\boldsymbol{\beta}}_{a_t}\right\rangle + \left\langle\begin{bmatrix}\boldsymbol{x}_t\\\phi^\star(\boldsymbol{x}_t)\end{bmatrix}, \begin{bmatrix}\widetilde{\boldsymbol{\theta}}_{a_t} - \boldsymbol{\theta}_{a_t}^\star\\\widetilde{\boldsymbol{\beta}}_{a_t} - \boldsymbol{\beta}_{a_t}^\star\end{bmatrix}\right\rangle\\
&\overset{(b)}{\leq} \left\|\widetilde{\phi}_t(\boldsymbol{x}_t) - \phi^\star(\boldsymbol{x}_t)\right\| \cdot \left\|\widetilde{\boldsymbol{\beta}}_{a_t}\right\| + \left\langle\begin{bmatrix}\boldsymbol{x}_t\\\phi^\star(\boldsymbol{x}_t)\end{bmatrix}, \begin{bmatrix}\widetilde{\boldsymbol{\theta}}_{a_t} - \boldsymbol{\theta}_{a_t}^\star\\\widetilde{\boldsymbol{\beta}}_{a_t} - \boldsymbol{\beta}_{a_t}^\star\end{bmatrix}\right\rangle\\
&\overset{(c)}{\leq} \left\|\widetilde{\phi}_t(\boldsymbol{x}_t) - \phi^\star(\boldsymbol{x}_t)\right\| \cdot \left\|\widetilde{\boldsymbol{\beta}}_{a_t}\right\| + \left\|\begin{bmatrix}\boldsymbol{x}_t\\\phi^\star(\boldsymbol{x}_t)\end{bmatrix}\right\|_{\boldsymbol{A}_{t-1,a_t}^{-1}}\left\|\begin{bmatrix}\widetilde{\boldsymbol{\theta}}_{a_t} - \boldsymbol{\theta}_{a_t}^\star\\\widetilde{\boldsymbol{\beta}}_{a_t} - \boldsymbol{\beta}_{a_t}^\star\end{bmatrix}\right\|_{\boldsymbol{A}_{t-1,a_t}}.
\end{aligned}
$$

where the inequality (a) is due to the definition of UCB, and (b) and (c) are obtained using the Cauchy-Schwarz inequality. Therefore, the cumulative regret can be further upper bounded by

$$
\begin{aligned}
R_T = \sum_{t=1}^{T}\Delta_t &\leq \sqrt{T\sum_{t=1}^{T}\Delta_t^2}\\
&\leq \sqrt{T\sum_{t=1}^{T}\left(\left\|\widetilde{\phi}_t(\boldsymbol{x}_t) - \phi^\star(\boldsymbol{x}_t)\right\|\left\|\widetilde{\boldsymbol{\beta}}_{a_t}\right\| + \left\|\begin{bmatrix}\boldsymbol{x}_t\\\phi^\star(\boldsymbol{x}_t)\end{bmatrix}\right\|_{\boldsymbol{A}_{t-1,a_t}^{-1}}\left\|\begin{bmatrix}\widetilde{\boldsymbol{\theta}}_{a_t} - \boldsymbol{\theta}_{a_t}\\\widetilde{\boldsymbol{\beta}}_{a_t} - \boldsymbol{\beta}_{a_t}\end{bmatrix}\right\|_{\boldsymbol{A}_{t-1,a_t}}\right)^2}\\
&\leq \sqrt{T\left(\sum_{t=1}^{T}2\left\|\widetilde{\phi}_t(\boldsymbol{x}_t) - \phi^\star(\boldsymbol{x}_t)\right\|^2\left\|\widetilde{\boldsymbol{\beta}}_{a_t}\right\|^2 + 2\zeta_T^2\left(1 \wedge \left\|\begin{bmatrix}\boldsymbol{x}_t\\\phi^\star(\boldsymbol{x}_t)\end{bmatrix}\right\|_{\boldsymbol{A}_{t-1,a_t}^{-1}}^2\right)\right)}\\
&\leq \sqrt{T\left(\sum_{t=1}^{T}2\left\|\widetilde{\phi}_t(\boldsymbol{x}_t) - \phi^\star(\boldsymbol{x}_t)\right\|^2\left\|\widetilde{\boldsymbol{\beta}}_{a_t}\right\|^2 + 2\zeta_T^2\left(1 \wedge \left\|\begin{bmatrix}\boldsymbol{x}_t\\\phi^\star(\boldsymbol{x}_t)\end{bmatrix}\right\|_{\boldsymbol{A}_{t-1,a_t}^{-1}}^2\right)\right)}
\end{aligned}
$$

In the next, we bound each term separately. Firstly, we have the following hold with probability at least $1 - \delta$ by using the union bound,

$$\sum_{t=1}^{T}2\left\|\widetilde{\phi}_t(\boldsymbol{x}_t) - \phi^\star(\boldsymbol{x}_t)\right\|^2 \cdot \left\|\widetilde{\boldsymbol{\beta}}_{a_t}\right\|^2 \leq 8\sum_{t=1}^{T}\left(e_t^{\delta/T}\right)^2.$$

In the next, to bound the remaining term, we use the result from Lemma 1. We first group the sums based the arm,

$$\sum_{t=1}^{T} 1 \wedge \left\| \begin{bmatrix} \boldsymbol{x}_t \\ \phi^\star(\boldsymbol{x}_t) \end{bmatrix} \right\|_{\boldsymbol{A}_{t-1,a_t}^{-1}}^2 = \sum_{a \in \mathcal{A}} \sum_{t \in [T]: a_t = a} 1 \wedge \left\| \begin{bmatrix} \boldsymbol{x}_t \\ \phi^\star(\boldsymbol{x}_t) \end{bmatrix} \right\|_{\boldsymbol{A}_{t-1,a}^{-1}}^2 \tag{16}$$

By denoting $n_T(a)$ as the number of times that arm $a$ is pulled, we can divide the arms into two groups,

$$\mathcal{G}_0 := \{a \in \mathcal{A} : n_T(a) = \Omega(\log(1/\delta))\} \quad \text{and} \quad \mathcal{G}_1 := \mathcal{A} \setminus \mathcal{G}_0.$$

Then, we can further decompose the r.h.s term of Equation 16 based on if the corresponding arm is in $\mathcal{G}_0$ or $\mathcal{G}_1$. Then, by applying Lemma 1, we have the following holds, with probability at least $1 - \delta$,

$$\sum_{a \in \mathcal{A}} \sum_{t \in [T]: a_t = a} 1 \wedge \left\| \begin{bmatrix} \boldsymbol{x}_t \\ \phi^\star(\boldsymbol{x}_t) \end{bmatrix} \right\|_{\boldsymbol{A}_{t-1,a}^{-1}}^2$$

$$= \sum_{a \in \mathcal{G}_0} \sum_{t \in [T]: a_t = a} 1 \wedge \left\| \begin{bmatrix} \boldsymbol{x}_t \\ \phi^\star(\boldsymbol{x}_t) \end{bmatrix} \right\|_{\boldsymbol{A}_{t-1,a}^{-1}}^2 + \sum_{a \in \mathcal{G}_1} \sum_{t \in [T]: a_t = a} 1 \wedge \left\| \begin{bmatrix} \boldsymbol{x}_t \\ \phi^\star(\boldsymbol{x}_t) \end{bmatrix} \right\|_{\boldsymbol{A}_{t-1,a}^{-1}}^2$$

$$\leq 2K d_u \log\left(1 + \frac{TL_u^2}{\lambda d_u}\right) + \frac{8K L_\epsilon^2 (L_\epsilon + L_x)^2}{\sigma_\epsilon^4} \log\left(\frac{32 K d_u L_\epsilon^2 (L_\epsilon + L_x)^2}{\delta \sigma_\epsilon^4}\right).$$

where the last inequality is due to Lemma 1 and apply the union bound on the $K$ arms. To bound the remainder term, since $\alpha \in [0, 1/2]$, by the learnability assumption as stated in Assumption 1 and Lemma 1, we have

$$\sum_{t=1}^{T} \left(e_t^{\delta/t}\right)^2 \leq \sum_{t=1}^{T} C_0^2 \cdot \left(1 \wedge \|\boldsymbol{x}\|_{\boldsymbol{X}_{t-1}^{-1}}^2\right)^{2\alpha} \cdot \log^2\left(\frac{tT}{\delta}\right)$$

$$\leq 4 C_0^2 T^{1-2\alpha} \log^{2\alpha}\left(\frac{\det \boldsymbol{X}_T}{\det \boldsymbol{X}_0}\right) \log^2\left(\frac{T}{\delta}\right)$$

$$\leq 4 C_0^2 T^{1-2\alpha} d_u^{2\alpha} \log^{2\alpha}\left(\frac{TL_u^2/d + \lambda}{\lambda}\right) \log^2\left(\frac{T}{\delta}\right)$$

Therefore, the total regret bound is bounded by the following term with probability at least $1 - 2\delta$,

$$\sqrt{T \cdot \left(8 C_0 T^{1-2\alpha} \log^{2\alpha}\left(\frac{\det \boldsymbol{X}_T}{\det \boldsymbol{X}_0}\right) \log^2\left(\frac{T}{\delta}\right) + K \zeta_T^2 \left(d_u \log\left(1 + \frac{TL_u^2}{\lambda d_u}\right) + \frac{48 L_\epsilon^4 L_u^2}{\sigma_\epsilon^4} \log\left(\frac{192 K d_u L_\epsilon^4 L_u^2}{\delta \sigma_\epsilon^4}\right)\right)\right)}$$

By hiding the logarithmic terms, we can further simplify it to be

$$\widetilde{\mathcal{O}}\left(T^{1-\alpha} d_u^\alpha + d_u \sqrt{TK}\right)$$

$\square$

## C.3. Missing Proofs in Generalizations

**Proposition 2.** *The regret of poLinUCB in Algorithm 1 for action-dependent contexts is upper bounded by* $\widetilde{\mathcal{O}}\left(T^{1-\alpha} d_u^\alpha \sqrt{K} + d_u \sqrt{TK}\right)$ *with probability at least* $1 - \delta$ *if* $T = \Omega(\log(1/\delta))$.

*Proof.* Our proof follows from the proof of Theorem 1. The immediate regret at each time step $t$ is

$$\Delta_t = r_{t,a_t^\star} - r_{t,a_t} \tag{17}$$

Recall that the definition of $a_t^\star$,

$$a_t^\star := \arg\max_{a \in \mathcal{A}} \langle \boldsymbol{\theta}_a^\star, \boldsymbol{x}_{t,a} \rangle + \langle \boldsymbol{\beta}_a^\star, \phi_a^\star(\boldsymbol{x}_{t,a}) \rangle. \tag{18}$$

To bound the immediate regret, we have

$$\Delta_t = \langle \boldsymbol{\theta}_{a_t^\star}^\star, \boldsymbol{x}_{t,a_t^\star} \rangle + \langle \boldsymbol{\beta}_{a_t^\star}^\star, \phi_{a_t^\star}^\star(\boldsymbol{x}_{t,a_t^\star}) \rangle - \langle \boldsymbol{\theta}_{a_t}^\star, \boldsymbol{x}_{t,a_t} \rangle - \langle \boldsymbol{\beta}_{a_t}^\star, \phi_{a_t}^\star(\boldsymbol{x}_{t,a_t}) \rangle. \tag{19}$$

By the definition of UCB, we further have

$$\Delta_t \leq \langle \widetilde{\boldsymbol{\theta}}_{t,a_t}, \boldsymbol{x}_{t,a_t} \rangle + \langle \widetilde{\boldsymbol{\beta}}_{t,a_t}, \widetilde{\phi}_{t,a_t}^\star(\boldsymbol{x}_{t,a_t}) \rangle - \langle \boldsymbol{\theta}_{a_t}^\star, \boldsymbol{x}_{t,a_t} \rangle - \langle \boldsymbol{\beta}_{a_t}^\star, \phi_{a_t}^\star(\boldsymbol{x}_{t,a_t}) \rangle. \tag{20}$$

By rearranging the terms, we get

$$\Delta_t \leq \underbrace{\langle \widetilde{\boldsymbol{\theta}}_{t,a_t} - \boldsymbol{\theta}_{a_t}^\star, \boldsymbol{x}_{t,a_t} \rangle + \langle \widetilde{\boldsymbol{\beta}}_{t,a_t} - \boldsymbol{\beta}_{a_t}^\star, \phi_{a_t}^\star(\boldsymbol{x}_{t,a_t}) \rangle}_{\textcircled{1}} + \underbrace{\langle \widetilde{\boldsymbol{\beta}}_{t,a_t}, \widetilde{\phi}_{t,a_t}(\boldsymbol{x}_{t,a_t}) - \phi_{a_t}^\star(\boldsymbol{x}_{t,a_t}) \rangle}_{\textcircled{2}} \tag{21}$$

Bounding the first term $\textcircled{1}$ is the same as the proof in Theorem 1, while bounding the second term $\textcircled{2}$ will be slightly different, as we now have $K$ such functions of $\phi_a^\star(\cdot)$ for $a \in \mathcal{A}$ to learn. By denoting the error for each estimate of $\phi_a^\star(\cdot)$ at iteration $t$ as $e_{t,a}^\delta$. Therefore, the contribution from the second term to the total regret can be bounded by

$$\sum_{t=1}^T \left( e_{t,a_t}^{\delta/t} \right)^2 = \sum_{a \in \mathcal{A}} \sum_{t \in [T]: a_t = a} \left( e_{t,a}^{t/\delta} \right)^2 \tag{22}$$

$$\leq 4KC_0 T^{1-2\alpha} \log^{2\alpha} \left( \frac{\det \boldsymbol{X}_T}{\det \boldsymbol{X}_0} \right) \log^2 \left( \frac{T}{\delta} \right). \tag{23}$$

Hence, by following the remaining steps in the proof of Theorem 1, we can conclude that the regret is upper bounded by

$$\widetilde{\mathcal{O}} \left( T^{1-\alpha} d_u^\alpha \sqrt{K} + d_u \sqrt{TK} \right), \tag{24}$$

where the only difference is the additional $\sqrt{K}$ appeared in the first term. $\qquad \square$

**Proposition 3.** *The regret of poLinUCB in Algorithm 1 for the above setting is upper bounded by $\widetilde{\mathcal{O}} \left( T^{1-\alpha} d_u^\alpha + d_u \sqrt{T} \right)$ with probability at least $1 - \delta$ if $T = \Omega(\log(1/\delta))$.*

*Proof.* This proof also follows from the proof of Theorem 1. The immediate regret at each time step $t$ is

$$\Delta_t = r_{t,\boldsymbol{x}_t^\star} - r_{t,\boldsymbol{x}_t} \tag{25}$$

Recall that the definition of $\boldsymbol{x}_t^\star$,

$$\boldsymbol{x}_t^\star := \arg\max_{\boldsymbol{x} \in D_t} \langle \boldsymbol{\theta}^\star, \boldsymbol{x} \rangle + \langle \boldsymbol{\beta}^\star, \phi^\star(\boldsymbol{x}) \rangle. \tag{26}$$

To bound the immediate regret, we have

$$\Delta_t = \langle \boldsymbol{\theta}^\star, \boldsymbol{x}_t^\star \rangle + \langle \boldsymbol{\beta}^\star, \phi^\star(\boldsymbol{x}_t^\star) \rangle - \langle \boldsymbol{\theta}^\star, \boldsymbol{x}_t \rangle - \langle \boldsymbol{\beta}^\star, \phi^\star(\boldsymbol{x}_t) \rangle \tag{27}$$

By the definition of UCB, we further have

$$\Delta_t \leq \langle \widetilde{\boldsymbol{\theta}}_t, \boldsymbol{x}_t \rangle + \langle \widetilde{\boldsymbol{\beta}}_t, \widetilde{\phi}_t^\star(\boldsymbol{x}_t) \rangle - \langle \boldsymbol{\theta}^\star, \boldsymbol{x}_t \rangle - \langle \boldsymbol{\beta}^\star, \phi^\star(\boldsymbol{x}_t) \rangle. \tag{28}$$

By rearranging the terms, we get

$$\Delta_t \leq \underbrace{\langle \widetilde{\boldsymbol{\theta}}_t - \boldsymbol{\theta}^\star, \boldsymbol{x}_t \rangle + \langle \widetilde{\boldsymbol{\beta}}_t - \boldsymbol{\beta}^\star, \phi^\star(\boldsymbol{x}_t) \rangle}_{\textcircled{1}} + \underbrace{\langle \widetilde{\boldsymbol{\beta}}_t, \widetilde{\phi}_t(\boldsymbol{x}_t) - \phi^\star(\boldsymbol{x}_t) \rangle}_{\textcircled{2}} \tag{29}$$

Since we only need to fit a single $\boldsymbol{\theta}^\star$, $\boldsymbol{\beta}^\star$ and $\phi^\star(\cdot)$. We thus have the following bound for the total regret,

$$\widetilde{\mathcal{O}} \left( T^{1-\alpha} d_u^\alpha + d_u \sqrt{T} \right). \tag{30}$$

$\square$

# D. Technical Lemmas

**Lemma 3** (**Confidence Ellipsoid**, based on Theorem 2 of (Abbasi-Yadkori et al., 2011)). *Let $\boldsymbol{w}^\star \in \mathbb{R}^d$, $\boldsymbol{V}_0 = \lambda \boldsymbol{I}$, $\lambda > 0$. For any $t \geq 0$, let $\boldsymbol{u}_1, \cdots, \boldsymbol{u}_t \in \mathbb{R}^d$, define $r_t = \langle \boldsymbol{u}_t, \boldsymbol{w}^\star \rangle + \eta_t$ where $\eta_t$ is $R_\eta$-sub-Gaussian and assume that $\|\boldsymbol{w}^\star\|_2 \leq L_{\boldsymbol{w}}$; let $\boldsymbol{V}_t = \boldsymbol{V}_0 + \sum_{s=1}^t \boldsymbol{u}_s \boldsymbol{u}_s^\top$ and $\hat{\boldsymbol{w}}_t$ be the corresponding regularised least-square estimator. Then, for any $\delta > 0$ and $t \geq 0$, with probability at least $1 - \delta$, $\boldsymbol{w}^\star$ lies in the set:*

$$\mathcal{C}_t = \left\{ \boldsymbol{w} \in \mathbb{R}^d : \|\hat{\boldsymbol{w}}_t - \boldsymbol{w}\|_{\boldsymbol{V}_t} \leq \sqrt{\lambda} L_{\boldsymbol{w}} + R_\eta \sqrt{2 \log \left( \frac{\det\left(\boldsymbol{V}_t\right)^{1/2} \det(\lambda \boldsymbol{I})^{-1/2}}{\delta} \right)} \right\}. \tag{31}$$

*Furthermore, if for all $t \geq 1$, $\|\boldsymbol{u}_t\| \leq L_{\boldsymbol{u}}$, then for any $\delta > 0$ and $t \geq 0$, with probability at least $1 - \delta$, $\boldsymbol{w}^\star$ lies in the set:*

$$\mathcal{C}_t = \left\{ \boldsymbol{w} \in \mathbb{R}^d : \|\hat{\boldsymbol{w}}_t - \boldsymbol{w}\|_{\boldsymbol{V}_t} \leq \sqrt{\lambda} L_{\boldsymbol{w}} + R_\eta \sqrt{d \log \left( \frac{1 + t L_{\boldsymbol{u}}^2 / \lambda}{\delta} \right)} \right\}. \tag{32}$$

**Lemma 4** (Bernstein's Inequality for Matrices, Theorem 6.1.1 of (Tropp et al., 2015)). *Let $\boldsymbol{X}_1, \cdots, \boldsymbol{X}_n \in \mathbb{R}^{d_1 \times d_2}$ be independent and centered random matrices. Assume that for each $i \in [n]$, $\boldsymbol{X}_i$ is* uniformly bounded*, that is:*

$$\mathbb{E}\left[\boldsymbol{X}_i\right] = \boldsymbol{0} \quad and \quad \|\boldsymbol{X}_i\| \leq B, \tag{33}$$

*where $\| \cdot \|$ denotes the spectral-norm distance here. Introduce the sum*

$$\boldsymbol{Z} = \sum_{i=1}^n \boldsymbol{X}_i, \tag{34}$$

*and let $\mathbb{V}(\boldsymbol{Z})$ denote the matrix variance statistics of the sum $\boldsymbol{Z}$:*

$$\mathbb{V}(\boldsymbol{Z}) = \max\left\{ \|\mathbb{E}\left[\boldsymbol{Z}\boldsymbol{Z}^*\right]\|, \|\mathbb{E}\left[\boldsymbol{Z}^*\boldsymbol{Z}\right]\| \right\} \tag{35}$$

$$= \max\left\{ \left\|\sum_{i=1}^n \boldsymbol{X}_i \boldsymbol{X}_i^*\right\|, \left\|\sum_{i=1}^n \boldsymbol{X}_i^* \boldsymbol{X}_i\right\| \right\}, \tag{36}$$

*where the asterisk $^*$ denotes the conjugate transpose operation. Then, for every $\epsilon \geq 0$, we have,*

$$\mathbb{P}\left(\|\boldsymbol{Z}\| \geq \epsilon\right) \leq (d_1 + d_2) \cdot \exp\left( \frac{-\epsilon^2/2}{\mathbb{V}(\boldsymbol{Z}) + B\epsilon/3} \right). \tag{37}$$

