# OpenReview forum: "Follow-ups Also Matter: Improving Contextual Bandits via Post-serving Contexts"
_ICML.cc/2023/Workshop/ILHF — ILHF Workshop ICML 2023_

### Official Review · Reviewer_UqP7 · 2023-06-14

**Rating:** 7
**Confidence:** 3

**Review:**

The authors consider a linear contextual bandit where part of the context is revealed to the learner after the action is taken. Under the assumption that this part of the context is predictable from the other part, they propose learning a predictor via ERM and then running a standard linUCB-style algorithm. To account for the multiple moving parts, they derive a novel extension of the standard elliptical potential lemma.

Overall, this paper makes a minor theoretical contribution to the space of linear contextual bandits. I appreciate the fact that experiments were performed using deep networks. Thematically, it is a great fit for this workshop.

Nitpicks: there are typos in Assumption 1 and Lemma 1 -- please fix them.

---

### Official Review · Reviewer_ezNh · 2023-06-16
**Review for submission 38**

**Rating:** 7
**Confidence:** 3

**Review:**


The paper is clear and easy to follow. The authors propose an linear bandit algorithm for online bandit when there are additional unobserved information which can be inferred from observed information through a learning algorithm.

Pros:
The paper offers valid theoretical and empirical analysis of their proposed algorithms. The problem setup is realistic and usual Linear Bandit may be suboptimal when additional information cannot be inferred by observed context through a linear function.

Cons:

While the analysis is interesting, this problem may not need a learning algorithm when going beyond linear bandit. If the regression oracle y|x,a is powerful enough, I don't see the need of an additional learning algorithm to regress z|x.

Minor: typos: "sthe standard EPL"; No reference on Lemma 1.

In the offline bandit setting, this setup is also referred to as runtime confounding [1].

[1] Coston, Amanda, Edward Kennedy, and Alexandra Chouldechova. "Counterfactual predictions under runtime confounding." Advances in neural information processing systems 33 (2020): 4150-4162.

---

### Decision · Program_Chairs · 2023-06-20

Accept